# PUO-Bench: A Panel Understanding and Operation Benchmark with A Privacy-Preserving Framework

**Wei Lin**[12*], **Yiwei Zhou**[13*], **Junkai Zhang**[14*], **Rui Shao**[15*], **Zhiyuan Zhao**[1],
**Junyu Gao**[15†], **Antoni B. Chan**[2], **Xuelong Li**[1†]

[1]The Institute of Artificial Intelligence, China Telecom (TeleAI)
[2]Dept. of Computer Science, City University of Hong Kong,  [3]Beijing Institute of Technology,
[4]Beijing University of Technology,  [5]Northwest Polytechnical University

## Abstract

Recent advancements in Vision-Language Models (VLMs) have enabled GUI agents to leverage visual features for interface understanding and operation in the digital world. However, limited research has addressed the interpretation and interaction with control panels in real-world settings. To bridge this gap, we propose the Panel Understanding and Operation (PUO) benchmark, comprising annotated panel images from appliances and associated vision-language instruction pairs. Experimental results on the benchmark demonstrate significant performance disparities between zero-shot and fine-tuned VLMs, revealing the lack of PUO-specific capabilities in existing language models. Furthermore, we introduce a Privacy-Preserving Framework (PPF) to address privacy concerns in cloud-based panel parsing and reasoning. PPF employs a dual-stage architecture, performing panel understanding on edge devices while delegating complex reasoning to cloud-based LLMs. Although this design introduces a performance trade-off due to edge model limitations, it eliminates the transmission of raw visual data, thereby mitigating privacy risks. Overall, this work provides foundational resources and methodologies for advancing interactive human-machine systems and robotic field in panel-centric applications.

## 1 Introduction

In recent years, the advancement of GUI agents in digital world has been remarkable, driven by significant improvements in both UI element parsing capabilities [65, 38, 53, 19, 40], the evolution of large language models (LLMs) [7, 24, 12, 3, 41] and Large Vision Language Models (VLMs) [4, 3, 43, 39, 44, 32, 8]. These agents are intelligent systems designed to interact with graphical user interfaces autonomously or semi-autonomously, enabling them to perform tasks such as navigating menus, filling out forms, clicking buttons, and extracting relevant information from UI components. Related algorithms also broadly benefit across interface adaptation [9, 14, 25], GUI-searches [16, 18, 36], and UI code generation [10, 15, 20, 62, 46].

While LLMs and VLMs have greatly advanced UI agent capabilities, their application remains largely limited to digital interfaces such as software[58, 56], websites [23], and mobile apps [59, 49]. These agents excel at interpreting screenshots or layouts and performing tasks based on instructions. However, if we broaden the scope of input sources to include vision-related data from the physical world, the potential applications extend far beyond GUI understanding and operation.

---

*This work was done by the starred authors during their internship at TeleAI, with equal contribution.
†Corresponding authors.

39th Conference on Neural Information Processing Systems (NeurIPS 2025) Track on Datasets and Benchmarks.

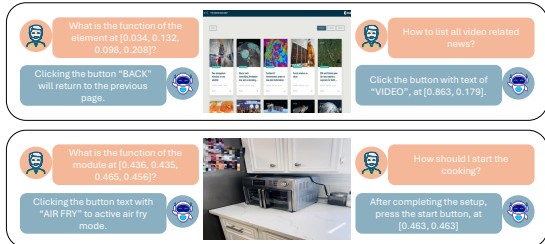

Figure 1: Comparison of UI understanding and operation in the digital world (top) vs. the real world (bottom), where captured images may contain private information, such as photos on walls (mosaic part).

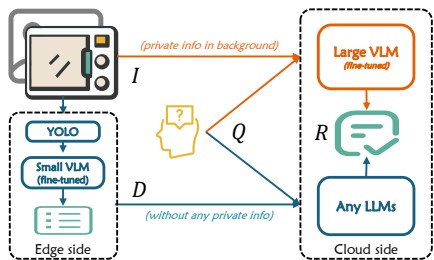

Figure 2: Information flow in the fine-tuned VLM (orange flow) *vs.* the proposed privacy-preserving framework with any cloud-side LLMs (blue flow).

As shown in Fig. 1, this shift opens up possibilities for deploying VLM-powered agents in real-world scenarios, where they can assist users in interacting with physical objects and environments featuring *panel interfaces*. Specifically, a VLM-powered agent with a camera can help users operate unfamiliar appliances, by parsing control panels, translating text, explaining symbols, and offering step-by-step instructions. This capability is beneficial in situations where one encounters unfamiliar physical interfaces and needs to interact with them in daily life. For example, travelers can rely on such agents to operate public devices, such as ticket machines or vending systems, making it easier for them to adapt to unfamiliar environments and enhancing their overall experience. Besides, it can also help workers interpret complex industrial equipment, and enable medical professionals to quickly familiarize themselves with new devices. Furthermore, a VLM capable of understanding panels holds significant potential for advancing general-purpose assistive robots by enhancing their ability to interpret and operate real-world appliances and equipment.

To advance real-world Panel Understanding and Operation (PUO), we introduce a large-scale image-based dataset containing over 19k annotated panel images and 430k instruction-following QA pairs. Using this dataset, we evaluate recent VLMs and fine-tune them to perform key PUO tasks: panel description, element grounding, function estimation, and goal-based planning. Our experiments show that current models struggle with panel-related knowledge, but fine-tuning on the PUO dataset significantly improves performance, demonstrating its value in bridging this gap.

However, we identify significant privacy concerns associated with deploying large VLMs in cloud environments for PUO. As illustrated in Fig. 1, unlike standardized digital interfaces that are uniform for all users, real-world images often contain sensitive details specific to a user's private environment. When users photograph their panels to request assistance, they may inadvertently capture personal or contextual information, which is subsequently transmitted to the cloud. This transmission poses a risk of exposing private data, rendering such an approach unsuitable for real-world applications where privacy is paramount. Especially in the future, it is unreasonable for robots to upload images of private spaces to the cloud solely for the purpose of understanding panels.

To address these privacy challenges, we propose a Privacy-Preserving Framework (PPF) for panel understanding and operation by integrating computation on the edge and cloud side, as shown in Fig. 2. This framework contains a compact VLM deployed on the edge device and a large LLM in the cloud side. The former extracts only the essential information about operable elements from the panel-related image, excluding any environmental or private data, while the latter can understand the parsed panel information and users' instructions, and then output instructive response to users. Besides, the edge-side model is required to output in natural language, allowing users to review and verify the sanitized content before it is transmitted to the cloud, ensuring no private information is included. Furthermore, any LLM can be deployed in the cloud to boost the flexibility of PPF.

In summary, our study makes three key contributions:

- We introduce a novel dataset for PUO. It comprises a diverse collection of panel images and a large set of instruction-following question-answer pairs, establishing a robust benchmark for PUO research.

- We propose the PPF for PUO, which ensures privacy by transmitting only sanitized panel information from edge devices to a cloud-based LLM. This approach eliminates the need to upload raw images, mitigating privacy risks associated with real-world image capture.
- Experiments on several VLMs reveal that current zero-shot VLMs lack the capability to understand panels, whereas fine-tuned models is able to address PUO tasks effectively. This highlights the absence of panel-related knowledge in existing models and underscores the importance of our proposed dataset.
- Experiments on PPF demonstrate its comparable ability to fine-tuned VLMs in element parsing and grounding, showcasing its potential to address PUO tasks in a privacy-preserving way.

## 2 Related work

While no prior work has specifically addressed panel understanding, we build on related research in UI understanding due to the conceptual similarities between control panels and GUIs.

### 2.1 UI understanding datasets

Over the past decade, numerous datasets have been developed to train and evaluate GUI agents across web, desktop, mobile, and cross-platform environments. For web-based agents, datasets like MiniWoB [30], MiniWoB++ [26], WebShop [27], Mind2Web [23], PhraseNode [48], and WebWalker [54] simulate interactions, map queries to actions, and enhance semantic understanding. In the desktop domain, OSWorld [58], ScreenAgent [47], Act2Cap [55], WindowsArena [11], SPider2-V [13], and AssistGUI [28] provide data on app behaviors, usability, and language-driven operations. For mobile interfaces, Rico [22], MoGUI & MoCon [70, 69], AIW [50], ScreenQA [34], AppSim & RefExp [6], and Meta-GUI [51] support research on app layouts, user interactions, and task-based scenarios. Cross-platform datasets like Screenspot [21] and GUIworld [17] further address diverse GUI challenges with screenshots, task instructions, and dynamic videos.

Together, these datasets provide a strong foundation for advancing GUI agents and enhancing human-computer interaction. However, they are largely limited to the digital world, overlooking real-world panel UIs in appliances. These real-world UIs have broader practical significance but remain underexplored. To fill this gap, we introduce the PUO dataset, extending UI understanding from digital interfaces to real-world environments.

### 2.2 UI understand & operation methods

Recent advancements in UI agents have been driven by two dominant paradigms: VLMs and LLM-powered frameworks, each contributing unique capabilities to enhance intelligent human-computer interaction. VLM-based approaches like CogAgent [33], SeeClick [21], and Pix2struct [38] excel in tasks such as fine-grained localization, GUI navigation, and structuring visual data into actionable insights. Meanwhile, LLM-powered methods, including SeeAct [66], WebVoyager [31], and AppAgent [63, 1], leverage multimodal reasoning, task decomposition, and hierarchical memory systems to enable seamless cross-platform interactions on web, mobile, and desktop environments. Together, these frameworks advance the state of automatic UI understanding and operation.

To explore the potential of UI understanding and operation in real-world scenarios, this paper introduces two algorithms: an end-to-end VLM model and an hybrid framework powered by LLM. These algorithms are designed to demonstrate the practical applicability and extensibility of PUO methodologies in complex, real-world environments.

## 3 PUO vision-language benchmark

### 3.1 Dataset overview

The PUO dataset is a vision-language instruction-following dataset designed to advance UI agent research by extending its scope from digital interfaces to real-world scenarios. While prior studies have primarily focused on digital UIs, our dataset introduces a diverse array of real-world control panels (Fig.3(a)), including those found on appliances such as air conditioners, washing machines,

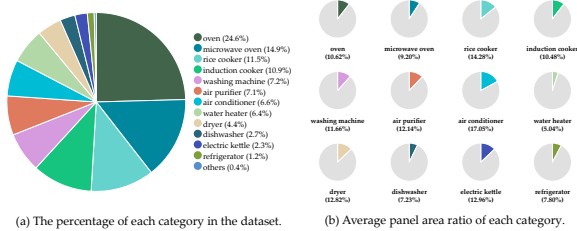

(a) The percentage of each category in the dataset.

(b) Average panel area ratio of each category.

Figure 3: Distribution of panel types and area ratio.

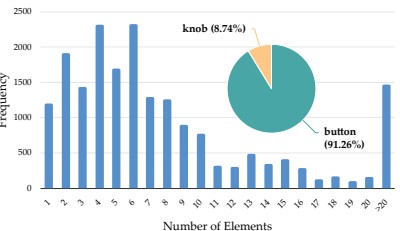

Figure 4: Histogram of element number.

and other devices. Unlike digital UIs, physical interfaces present unique challenges, such as image blur, perspective distortions, and complex backgrounds. Moreover, the panel typically occupies only a fraction of the image frame. As shown in Fig. 3(b), the area occupied by the panel across all categories is less than one-quarter of the image area, significantly smaller than that in digital UIs which are usually full screenshots encompassing 100% of the image content.

Our dataset contains 19,118 panel images with varying complexity, each containing 1 to 35 operable components. Fig. 4 present a histogram to visualize the component density in detail. Approximately 6.2% of the panels contain only a single component, while around 6.8% feature more than 20 components, with the remainder falling in between. This variation ensures that models trained on the dataset can handle both simple and highly intricate interfaces.

Additionally, similar to GUIs where most elements are clickable, most operable elements in real-world panels are buttons. However, some panels also include knobs. As shown in Fig. 4, the ratio of buttons to knobs is approximately 9:1, highlighting the dominance of buttons while underscoring that knobs serve as a secondary means of interaction in certain panel contexts.

## 3.2 Element annotation details

In the proposed PUO dataset, each operable UI element is annotated with five key attributes: *type*, *position*, *text*, *icon*, and *function*, as summarized in Tab. 1. The *type* attribute specifies the interaction mechanism of the element rather than its visual form. We classify all operable components into two categories, *buttons* and *knobs*, based on how users manipulate them. Buttons represent binary operations that toggle between two states (e.g., on/off, start/stop), while knobs represent multi-state controls that adjust a continuous or discrete value by position (e.g., temperature, speed). Accordingly, sliders, toggles, and tap-based switches are annotated as knobs, since their operational characteristics are equivalent to adjusting state through position rather than through distinct toggles. For touchscreens or dynamic displays, each touchable region is annotated as an independent button, consistent with conventions in GUI and UI understanding tasks.

The *position* attribute defines the bounding box of each operable element, covering not only the control itself but also relevant contextual cues such as labels or nearby symbols. For instance, in Fig. 5, the bounding box of a button includes the adjacent label "Temp," while that of a knob encloses text like "Temperature °F" and options such as "350°F" or "warm." These textual cues are recorded in the *text* attribute to convey semantic context, while graphical cues such as arrows or icons are stored in the *icon* attribute (e.g., an upward triangle indicating an increase in temperature). Finally, the *function* attribute provides a concise description of each element's operational role within the appliance. This annotation schema ensures consistent labeling across appliances and enables models to infer functionality by leveraging both local and contextual information, allowing trained VLMs to understand control elements in an explicit and interpretable manner.

As shown in Fig. 6, we also present the positional distribution and size of buttons and knobs across all images in our dataset. Fig. 6(a) shows that a significant number of buttons tend to cluster around the center of the panels, whereas knobs are more commonly located in the upper-right region(Fig. 6(b)). This pattern contrasts with the distribution of elements in GUIs, where clickable components are typically spread across the entire interface. Furthermore, in Fig. 6(b & d), we provide box plots illustrating the width and height distributions of the bounding boxes that enclose buttons and knobs. Notably, the dimensions of most knobs range from 0.01 to 0.33 of the total image size, whereas for buttons, both the width and height exhibit a third quartile value below 0.07. This indicates that buttons

Table 1: Annotation of elements in PUO dataset.

| Keyword | Explanation |
|---|---|
| Type | The category of the UI element, such as "button" or "knob," which defines its interaction style. |
| Position | Coordinates of the top-left and bottom-right corners of the UI element's bounding box. |
| Text | Surrounding textual information that provides context or describes the purpose of the UI element in detail. |
| Icon | Visual symbol or graphic associated with the UI element, offering additional context about its role or category. |
| Function | Specific operational role or task performed by the UI element on the appliance it belongs to. |

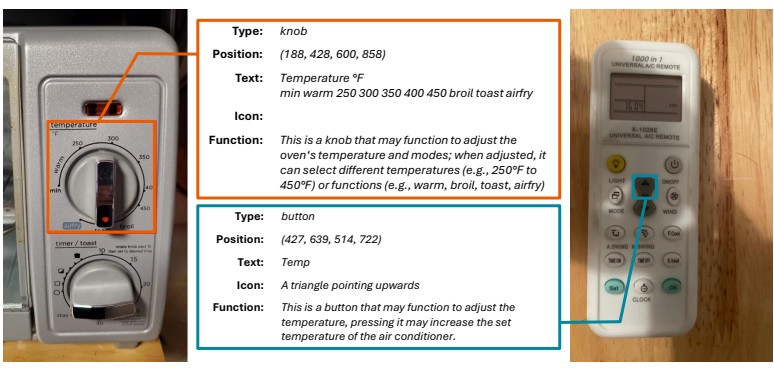

Figure 5: Two examples of annotated elements.

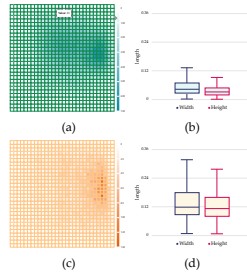

Figure 6: The location and size distributions of button (a & b) and knob (c & d).

are generally smaller and more compact compared to knobs, reinforcing the distinction between the two types of elements not only in their placement but also in their physical proportions.

## 3.3 Benchmark Tasks

To systematically evaluate the capability of models to interpret and interact with real-world interfaces, the proposed benchmark introduces four tasks assessing both holistic comprehension and fine-grained reasoning. These tasks are structured to address distinct challenges in PUO, as detailed below:

**Panel Description:** This task evaluates a model's ability to generate a structured natural language description of the entire panel interface. The model must enumerate all operable components, characterize their functional attributes. This ensures comprehensive visual-semantic alignment between panel elements and their operational purposes.

**Element Grounding:** Focused on spatial-semantic mapping, this task requires the model to localize operable elements corresponding to a specified goal. Given a high-level objective, the model must output precise 2D coordinates of the relevant element while ensuring alignment between user intent and particular elements.

**Functional Inference:** Here, the model is required to analyze a specified element (via coordinates or scene context) to determine its purpose, operational constraints, and potential effects of interactions. This involves interpreting visual or symbolic cues and reasoning about their role within the interface.

**Multi-Step Operation Planning:** This task requires the model to plan a sequence of actions needed to achieve a specified goal by leveraging visual and contextual information. The model must identify operable elements on the panel, infer the system's current state from the input query, determine the necessary steps and their order, and link each action to the corresponding UI components to achieve the desired outcome. This process integrates visual analysis, logical reasoning, and contextual understanding to enable effective interaction with complex interfaces.

Fig. 7 illustrates the four benchmark tasks using a representative panel interface. For panel description , the model generates a structured summary of operable components (e.g., buttons, knobs) and their functional roles, such as power adjustment or parameter configuration. In element grounding , the model localizes a target element (e.g., a control associated with a predefined goal) by predicting its 2D coordinates. Functional inference requires deducing the purpose and operational impact of a specified element, such as interpreting a button's role in pausing processes. Finally, multi-step

**(a) Panel Description**

- *Button 'Micro Power' - This button may function to adjust the microwave power level.*
- *Button 'Express' - This button may function to provide quick cooking settings.*
- *Button 'Defrost/Clock' - This button may function to defrost food and set the clock.*
- *Button 'Auto Cooking' - This button may function to automatically set cooking parameters based on food type.*
- *Knob 'Time' and 'Weight' - This knob may function to set time and weight for cooking or defrosting.*
- *Button 'Pause/Cancel' - This button may function to pause or cancel the current operation.*
- *Button 'Instant/Start' - This button may function to start the cooking process immediately.*

**(b) Element Grounding**

- ***Question:*** *I want to thaw some frozen meat. Which control should I use for that?*
- ***Answer:*** *You should use the button labelled "Defrost/Clock" to activate the defrosting function. Therefore, click button at [0.743, 0.466].*

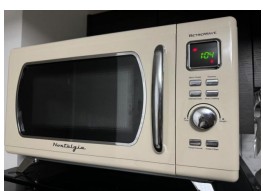

**(c) Function Inference**

- ***Question:*** *If the microwave is heating food, what happens when pressing the button located at [0.718, 0.708, 0.773, 0.767]?*
- ***Answer:*** *When the microwave oven is heating food and you press this button, it will pause the current heating operation. This allows you to temporarily stop the process without cancelling the settings or turning off the oven.*

**(d) Multi-Step Operation Planning**

- ***Question:*** *You have some frozen food and you want to defrost it first, then cook it using the Auto Cooking mode. How would you go about setting this on your microwave oven?*
- ***Answer:*** *To defrost the frozen food and then cook it using the Auto Cooking mode on your microwave oven, follow these steps:*
1. *Press the 'Defrost/Clock' button (labelled with 'Defrost/Clock') to activate the defrosting operation. This will set the microwave to defrost mode.*
2. *Using the 'Time - Weight +' knob to adjust according to the weight of the food and ensure proper defrosting.*
3. *Press the 'Pause/Cancel' button (labelled with 'Pause/Cancel') to stop the defrost operation once defrosting is complete.*
4. *Press the 'Auto Cooking' button (labelled with 'Auto Cooking') to select the automatic cooking mode.*
5. *Press the 'Instant/Start' button (labelled with 'Instant/Start') to start the automatic cooking mode.*

Figure 7: Illustration of the four benchmark tasks on a microwave oven panel, demonstrating panel description, element grounding, function inference, and multi-step operation planning.

operation planning involves synthesizing visual and contextual inputs to devise action sequences (e.g., defrosting followed by automated cooking mode activation). Collectively, this example demonstrates how the benchmark evaluates layered competencies – ranging from descriptive accuracy to reasoning in real-world interface interaction.

# 4 Methods for PUO tasks

This chapter introduces two methodological frameworks for panel understanding and operation. The first is an end-to-end system that uses a VLM to handle all tasks in the PUO benchmark, combining visual and textual processing in a unified workflow. The second, PPF, uses a modular approach: lightweight edge models first parse the panel, then send structured data and queries to a cloud-based LLM for responses. This design avoids uploading raw images, reducing privacy risks.

## 4.1 End-to-end VLMs

To develop an end-to-end VLM for PUO, we employ a methodology akin to digital UI agent frameworks, fine-tuning a VLM to operate seamlessly in an integrated manner. This approach jointly process visual and textual inputs, enabling the model to interpret control panels and generate contextually appropriate responses based on user instructions. Specifically, we utilize Low-Rank Adaptation (LoRA) [35] to fine-tune VLMs, which can balance efficient parameter updates with minimal computational overhead.

Denote the fine-tuned VLM as $\mathcal{V}$, which takes an image $I$ of the control panel and a textual instruction $Q$ as inputs and outputs a response $R$. LoRA approximates weight updates using low-rank matrices $\Delta W$, expressed as:

$$R = \mathcal{V}(I, Q; \Theta_v + \Delta W), \tag{1}$$

where $\Theta_v$ represents the pre-trained parameters of $\mathcal{V}$, and $\Delta W$ adopts a low-rank structure defined as $\Delta W = AB^T$. Here, $A \in \mathbb{R}^{d \times r}$ and $B \in \mathbb{R}^{r \times d}$ with $r \ll d$. LoRA significantly reduces trainable parameters, enabling computationally efficient fine-tuning while retaining the pre-trained knowledge of the base model [35], which allows the fine-tuned model to leverage its foundational capabilities while adapting to the PUO task's unique demands.

By comparing the performance of zero-shot and fine-tuned VLMs, we can assess the extent to which existing VLMs inherently understand panels in real world. A significant performance gap between zero-shot and fine-tuned models across tasks indicates that current VLM knowledge bases lack PUO-specific knowledge. This disparity underscores the critical necessity of our proposed PUO benchmark for advancing research and capabilities in real-world interface interaction.

# 5 Privacy-preserving framework for PUO

**Motivation.** When capturing images of physical control panels, sensitive environmental details are often inadvertently included, such as personal photos on walls, as shown in the bottom scene of Fig. 1. This can lead to unintended exposure of private data, especially when the capturing device is an autonomous robot that might upload such information to the cloud without the owner's awareness. To better quantify this risk, we analyzed real-world panel images and found that, as shown in Fig. 3(b), panels occupy less than 20% of the pixels on average, meaning that most of the captured content is unrelated background. We further used a vision-language model to automatically examine each image and identify potential privacy-sensitive elements (e.g., faces, names, phone numbers, or identifiable locations). The analysis reveals that approximately 3.67% of the images contain personal information, indicating a non-negligible privacy risk in everyday use. To address this issue, we propose the Privacy-Preserving Framework (PPF), a two-tier architecture that decouples edge-side panel parsing from cloud-based reasoning. In this design, raw images are processed locally and never transmitted externally, ensuring that sensitive contextual information remains private while still enabling intelligent cloud-side interaction.

**Design details.** To improve the accuracy of edge-side panel parsing, we design the PPF that combines a lightweight vision-language model ($\mathcal{E}$) with a dedicated vision-based detector to identify all operable elements in panel images. This integration is both effective and straightforward, as PPF's modular design naturally supports flexible component composition. The motivation for this enhancement stems from our initial experiments, where relying solely on $\mathcal{E}$ revealed performance limitations due to constrained model capacity and coarse-grained element detection.

Our implementation of PPF is shown in Fig. 2 (blue flow). YOLO [37] is trained to detect a set $B$ including all operable elements: $B = \{(c_i, b_i)\}_{i=1}^N$, where $c_i \in \{\text{button}, \text{knob}\}$ denotes the element type and $b_i = (x_1, y_1, x_2, y_2)$ defines its bounding box coordinates. Following it, the set $B$ are then passed to the edge-side parser $\mathcal{E}$ to predict a function description $f_i$ for each element $b_i$ in nature language. The edge-side model outputs a structured set of tuples $P = \{(b_i, f_i)\}_{i=1}^N$, defined as:

$$P = \mathcal{E}(I, B; \Theta_e), \tag{2}$$

where $\Theta_e$ denotes the parameters of the fine-tuned parser $\mathcal{E}$. Notably, the element class $c_i$ is excluded from $P$, as this information is already encapsulated within the $f_i$.

By integrating YOLO's detection results with $\mathcal{E}$, the edge model achieves robust panel parsing while preserving privacy. While YOLO introduces a little computational overhead, this trade-off is justified by its significant accuracy improvements. Importantly , the output $\mathcal{P}$ excludes environmental or private data and retains only task-relevant panel information.

Once $P$ is generated by the edge-side model, it is combined with the user's instruction $Q$ and transmitted to a cloud-based LLM to generate response $R$:

$$R = \mathcal{L}(Q, P; \Theta_l), \tag{3}$$

where $\mathcal{L}$ represents the LLM's inference process parameterized by $\Theta_l$.

Fig. 2 contrasts the workflows of end-to-end VLMs and PPF: while end-to-end systems process raw images directly in the cloud, PPF confines sensitive visual processing to the edge device, transmitting only structured panel data $P$ to the cloud, which ensures privacy preservation. Although PPF adopts a simple workflow by splitting computation between edge and cloud, it provides theoretical guarantees that private visual data never leaves the edge device, thereby eliminating the risk of data leakage.

# 6 Experiments

We evaluate model performance on the four PUO benchmark tasks. For end-to-end methods, we test Yi-VL(6.72B) [61], LLAVA-NeXT(7.57B) [44], and Qwen2.5-VL(8.29B) [8]. In PPF setups, we deploy the lightweight Qwen2.5-VL(3.75B) [8] on the edge, with GPT-4o [3], Qwen2.5-72B-instruction [60], and Claude [5] serving as the cloud-side LLM for (3). All models are fine-tuned via LLaMA Factory [68].

Table 2: Comparison on PUO benchmarks.

| Method | FT | Description | | | Grounding | | Function | | Planning |
|---|---|---|---|---|---|---|---|---|---|
| | | $C_{\text{CIDEr}}$ | $C_{\text{BERT-S}}$ | LJS-F1 | Coord | Acc | BERT-S | LJS-Acc | LJS-Acc |
| GPT-4o | ✗ | 35.40 | 0.769 | 0.545 | 0.992 | 0.063 | 0.900 | 0.297 | 0.294 |
| Claude-3.7 | ✗ | 30.31 | 0.765 | 0.538 | 0.985 | 0.072 | 0.894 | 0.274 | 0.192 |
| Qwen-VL-72B | ✗ | 0.094 | 0.117 | 0.464 | 0.941 | 0.082 | 0.894 | 0.265 | 0.122 |
| Yi-VL-6B [61] | ✗ | 5.34 | 0.455 | 0.068 | 0.000 | 0.000 | 0.091 | 0.167 | 0.030 |
| | ✓ | 163.45 | 0.802 | 0.527 | 0.992 | 0.106 | 0.910 | 0.290 | 0.393 |
| LLAVA-7B [42] | ✗ | 13.09 | 0.627 | 0.146 | 0.000 | 0.000 | 0.869 | 0.124 | 0.049 |
| | ✓ | **248.93** | 0.879 | **0.749** | 0.995 | 0.498 | 0.918 | 0.463 | **0.490** |
| Qwen-VL-7B [8] | ✗ | 23.86 | 0.658 | 0.408 | 0.000 | 0.000 | 0.876 | 0.349 | 0.239 |
| | ✓ | 238.23 | 0.884 | 0.748 | 0.995 | 0.625 | **0.918** | 0.528 | 0.444 |
| PPF *(w/ GPT-4o)* | ✓ | 192.90 | **0.898** | 0.706 | 0.990 | 0.673 | 0.903 | **0.630** | 0.420 |
| PPF *(w/ Claude)* | ✓ | 189.72 | 0.898 | 0.706 | 0.990 | **0.676** | 0.906 | 0.532 | 0.335 |
| PPF *(w/ Qwen)* | ✓ | 191.01 | 0.898 | 0.709 | **0.997** | 0.668 | 0.909 | 0.510 | 0.315 |

## 6.1 Panel description

The goal of panel description is to completely list all operable controls and their functions in the panel. To evaluate how well models achieve this goal, we extract predicted elements $\{e_i\}_{i=1}^{N_{\text{pred}}}$ and GT elements $\{e'_j\}_{j=1}^{N_{\text{gt}}}$, and construct a similarity matrix $S \in \mathbb{R}^{N_{\text{pred}} \times N_{\text{gt}}}$, where $S_{ij} = \text{sim}(e_i, e'_j)$ measures similarity using CIDEr [52] or BERT-S [64]. The Hungarian algorithm finds the optimal matching $M \in \{0, 1\}^{N_{\text{pred}} \times N_{\text{gt}}}$, yielding the score:

$$C_{\text{sim}} = 2\langle S, M\rangle/(N_{\text{pred}} + N_{\text{gt}}), \tag{4}$$

where $\langle \cdot, \cdot \rangle$ is the Frobenius inner product. We also propose LJS-F1 based on LLM-as-Judge [67]:

$$\text{LJS-F1} = 2N_{\text{match}}/(N_{\text{pred}} + N_{\text{gt}}), \tag{5}$$

with $N_{\text{match}}$ being the number of semantically matched elements judged by an LLM.

A detailed comparison is listed in the *Description* column of Tab. 2. Compared to zero-shot models, LoRA-fine-tuned models show significant improvement in identifying operable elements, indicating current VLMs lack panel-related knowledge. PPF with GPT-4o as cloud-side model outperforms other fully zero-shot models, achieving the highest $C_{\text{BERT-S}}$ score and third-highest LJS-F1, surpassing fine-tuned YI-VL [61] in all aspects.

## 6.2 Element grounding

The element grounding task evaluates a model's ability to associate textual descriptions with specific operable elements on a panel by predicting their spatial coordinates. We evaluate performance using two metrics: (1) *Coord*, the relative frequency of parseable coordinates in the response; and (2) *Acc*, the accuracy of predicted coordinates that fall within the ground-truth element's bounding box.

As shown in the *Grounding* column of Tab. 2, zero-shot models output invalid coordinates (Coord =0.000), while their fine-tuned versions and online large VLMs are able to produce valid coordinate information. However, the accuracy of online large VLMs is below 0.1. Notably, PPF with GPT-4o as the LLM achieves the highest localization accuracy (Acc = 0.673), surpassing all fine-tuned models, thanks to YOLO's robust detection capabilities.

## 6.3 Function inference

The function inference task evaluates a model's ability to deduce the functional properties of an element. The corresponding performance is measured using BERT-S [64] and LJS-Acc based on [67]. Differently, LLM here is to determine whether the predicted response matches the ground truth:

$$\text{LJS-Acc} = \frac{1}{N} \sum_{i=1}^{N} \text{LLM}_{pt}(R_i, R'_i), \tag{6}$$

Table 3: Ablation study on with and without detection results.

| Method | YOLO | Description | | | Grounding | | Function | | Planning |
|---|---|---|---|---|---|---|---|---|---|
| | | $C_{\text{CIDEr}}$ | $C_{\text{BERT-S}}$ | LJS-F1 | Coord | Acc. | BERT-S | LJS-Acc. | LJS-Acc. |
| PPF *(w/ GPT-4o)* | ✗ | 156.01 | 0.854 | 0.641 | 0.989 | 0.515 | 0.897 | 0.456 | 0.333 |
| | ✓ | **192.90** | **0.898** | **0.706** | **0.990** | **0.673** | **0.903** | **0.567** | **0.420** |
| Qwen-VL-7B [8] | ✗ | **238.23** | **0.884** | **0.748** | 0.995 | 0.625 | 0.918 | 0.528 | **0.444** |
| | ✓ | 186.86 | 0.875 | 0.711 | **0.998** | **0.705** | **0.956** | **0.551** | 0.416 |

where $\text{LLM}_P(R_i, R'_i) = 1$ if the response $R_i$ is semantically equivalent to the GT $R'_i$, as determined by the prompt $pt$; otherwise, it is 0.

Results in the *Function* column of Tab. 2 show that fine-tuned Qwen-VL performs best in functional reasoning. PPF ranks second in LJS-Acc, showing strong semantic alignment despite edge constraints. PUO involves bidirectional reasoning: inferring function from coordinates and localizing elements from descriptions – both essential for real-world use. Though PPF shows slight semantic gaps compared to fine-tuned models, its ability to parse and reason about panel elements highlights its practical value for privacy-sensitive applications.

### 6.4 Multi-Step operation planning

The multi-step operation planning task evaluates a model's ability to generate goal-driven action sequences. Since PUO is an image-based dataset, planning objectives are constrained to predefined scenarios in input questions, excluding real-time panel feedback from the task scope. Performance is assessed via (6) using specific prompt: whether the predicted plan aligns with the GT in operational logic and goal achievement.

The *Planning* column of Tab. 2 highlights key findings for this task. Fine-tuned LLAVA-NeXT achieves the highest planning accuracy, while Qwen-VL closely matches its performance. PPF shows significant improvement over zero-shot models but lags behind fine-tuned VLMs, reflecting limitations in LLM knowledge bases for PUO-specific planning.

### 6.5 Discussion on model with/without detection results

The comparative performance of model with and without YOLO integration is summarized in Tab. 3. In the latter case, the edge-side model directly parses panel elements from images:

$$P = \mathcal{E}'(I; \Theta_{e'}). \tag{7}$$

Although the lightweight edge model demonstrates baseline grounding capabilities independently, integrating YOLO's robust detection significantly amplifies PPF's efficacy across all evaluated tasks. However, embedding the results of detection into an end-to-end VLM reduces its own parsing and planning capabilities. Therefore, it is only recommended to introduce YOLO results into PPF to enhance the parsing ability of small edge-side models.

## 7 Conclusion

This study introduces Panel Understanding and Operation (PUO) , a novel task involving the extraction of information from real-world panel images and the generation of actionable suggestions based on user instructions. Despite its practical relevance and potential impact on robotics, PUO remains largely underexplored in vision-language research. To address this, we construct a comprehensive dataset with annotated images and diverse question-answer pairs, establishing a benchmark containing four core tasks for evaluating PUO performance.

Using this benchmark, we evaluate both zero-shot and fine-tuned VLMs, revealing significant limitations in current models. We also identify privacy concerns in PUO applications. To mitigate these, we propose a Privacy-Preserving Framework (PPF) that splits the pipeline into two stages: an edge-side detector-parser handles panel understanding and grounding locally, while only parsed

semantic data and instructions are sent to the cloud for response generation. Although PPF incurs some performance loss due to edge model constraints, it effectively avoids the exposure of raw visual data during transmission.

**Challenges:** Our findings reveal that existing VLMs lack prior knowledge of panel understanding and operation, resulting in a significant performance gap between fine-tuned and zero-shot models. Even when integrating LLMs into the cloud-side of PPF, notable discrepancies persist in planning and description tasks compared to fine-tuned VLMs. This highlights a critical limitation: current LLMs are not pre-trained on PUO-related knowledge. These results underscore the importance of the PUO benchmark in advancing computer vision and addressing real-world interaction challenges. The ability to ground operable components also remains underdeveloped in current models, yet is essential for effective visual reasoning and real-world panel interaction.

**Future works**: Future research on PUO can focus on developing hybrid architectures that better integrate edge computing with cloud-based intelligence, aiming to close the performance gap while maintaining privacy. Furthermore, expanding the benchmark to include dynamic panel interactions would more closely align PUO with real-world robotic applications. Building upon this foundation, integrating PUO with embodied agents capable of real-time robotic manipulation could enable seamless transitions between panel interpretation and physical action. Such advancements have the potential to significantly contribute to the field of robotics, fostering intelligent systems that can understand and interact with complex human-made interfaces in real environments.

## Acknowledgments

This work was supported in part by the Strategic Research Grant of City University of Hong Kong (Project No. 7005840) and in part by the National Natural Science Foundation of China (Grant No. 62306241).

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

- Sec. D reports the human evaluation of the LJS score and its variation across annotators;
- Sec. E describes how the edge-side parser is trained;
- Sec. F visualizes the differences between the zero-shot and fine-tuned model results.

# A  Annotation Guideline

To ensure annotation quality and consistency, the labeling process in PUO is organized as a **step-by-step pipeline**, rather than having a single annotator complete all five types of annotations simultaneously. Each step focuses on a specific attribute group, allowing cross-verification between annotators and tools to minimize subjective bias and improve reliability. The overall annotation workflow is summarized as follows.

**(1) Type and Position Annotation.**  This step resembles conventional object detection tasks. Annotators first label all operable elements in each panel image by specifying their *type* (i.e., button or knob) and corresponding *bounding boxes*. Each bounding box is drawn to cover not only the element itself but also relevant contextual information, such as adjacent texts or symbols (as illustrated in Fig. 5). This design provides the necessary spatial context for subsequent text, icon, and function annotation.

**(2) Text and Icon Annotation.**  Next, we extract textual and graphical cues from the annotated regions. To improve efficiency, we leverage existing vision-language models equipped with OCR and icon recognition capabilities to automatically obtain initial labels for texts and icons within each bounding box. These preliminary results are then manually verified and corrected through a custom-developed **web annotation tool**, which assists human annotators in identifying mislabeling and missing entries.

**(3) Function Annotation.**  Finally, we annotate the *functional role* of each operable element using a semi-automated process that combines human supervision with large language model (LLM) reasoning. A web interface connects to an LLM, which receives a structured prompt describing the product category, element type, surrounding text, and associated icon, as well as optional additional context filled by the annotator. The prompt takes the following form:

```
This is an operable element on a {product}, and its type is {type}.
The text written on or around this component is:  '{text}'.
The icon description associated with this component is:  '{icon}'.
Besides, its function is also related to '{other_prompts}'.
Based on these details, what function might this element serve in
operating the {product}?
```

In most cases, the LLM can correctly infer the element's function directly from contextual cues. When necessary, annotators supplement missing contextual hints via the {other_prompts} field to ensure accurate and consistent results.

Through this three-stage annotation pipeline, the *type*, *position*, and contextual cues (Steps 1 & 2) are grounded in actual image content, while the *function annotation* (Step 3) is standardized through structured prompting and human verification. This design guarantees stepwise consistency and reproducibility across all annotations.

## B  Training Details

During the training process, we fine-tuned all VLMs using LoRA on a combined dataset consisting of QA pairs from four distinct tasks to measure the model's performance on the PUO task.

The fine-tuning was conducted using LLaMA Factory [68], leveraging the SFT stage with a learning rate of 1e-4 for 10 epochs on a machine equipped with 8 A100 GPUs. We allowed the model to generate up to 4096 new tokens per response. To ensure diverse and natural responses, we set the temperature to 0.95 and applied Top-p sampling (nucleus sampling) with a p-value of 0.7.

## C  Details about LLM-as-Judge

In Section 6, we present two metrics for panel description, function estimation, and multi-step operation planning: LJS-F1 and LJS-Acc, as defined in (5) and (6), respectively. It is important to note that our approach differs from the vanilla method [67], which uses an LLM to judge the quality of outputs using continuous scores. In contrast, we employ discrete, binary scoring to assess whether the prediction matches the GT.

### C.1  LJS-F1 for panel description

For panel description, we use the prompt like this:

---

**The prompt for LJS-F1 in panel description**

*The following markdown content provides two descriptions of the same control panels on an appliance. According to these descriptions, output a number 'A' indicating how many elements are in the first description, a number 'B' indicating how many elements are in the second description, and a number 'C' indicating how many elements are in both the frist and the second description. The 'element' means a button or a knob.*
*"'md*
*{ ground truth }*
*"'*

*"'md*
*{ prediction }*

*"'*
*Analysis should be present. In the last line of the answer, outputing three numbers bounded with square brackets: "[A, B, C]", indicating element number in the first and the second description, and the number of common elements.*

---

In Fig. 8, we present an example to show how LLM judges the element count in ground truth and prediction, and how the number of common elements is estimated.

### C.2  LJS-Acc for function estimation

For function estimation, we only let the LLM to determine whether the prediction and ground truth describe the same element in the function level:

---

**The prompt for LJS-Acc in function estimation**

*The following markdown content provides two descriptions of two buttons or knobs on an appliance. Judge whether the two pieces of content describe the same button/knob and the same function.*
*"'md*
*{ ground truth }*
*"'*

---

**ground truth**

- Button 'UNIT' - This button relates to the operations of the device's units. Pressing it will switch unit settings or confirm unit operations.
- Button 'AVG/VPP' - This button is used to switch the display mode or change related parameters. When pressed, it can toggle between modes like AVG (Average) and VPP (Peak Value).
- Button 'HOLD' (Power icon) - This button controls the power switch of the air purifier. Pressing it will turn the device on or off.

**question**

Could you list all the functional modules within this appliance panel image and explain what each one does?

**prediction of ppf-claude**

- Button 'ON/OFF' (A power symbol) - This button controls the power switch of the detector. Pressing it turns the detector on or off.
- Button 'unit' - This button provides function options related to unit selection. Pressing it triggers corresponding operations.
- Button 'TEST' (A button with the word 'TEST') - This button activates the test function of the detector. Pressing it displays the test status of the detector.

**LLM-as-Judge**

To analyze the markdown content, we need to identify and count the buttons and knobs described in both sections.

**First Description Analysis:**
1. **Button 'UNIT'** - A button related to the operations of the device's units.
2. **Button 'AVG/VPP'** - A button used to switch the display mode between AVG and VPP.
3. **Button 'HOLD' (Power icon)** - A button to turn the device on or off.

The total number of elements in the first description is 3.

**Second Description Analysis:**
1. **Button 'ON/OFF' (A power symbol)** - A button to control the power switch of the detector.
2. **Button 'unit'** - A button to provide function options related to unit selection.
3. **Button 'TEST' (A button with the word 'TEST')** - A button to activate the test function of the detector.

The total number of elements in the second description is 3.

**Common Elements Analysis:**
Now we need to identify any common elements between the two descriptions. We focus on the names and descriptions which are related in their functionality.
1. The function of button 'UNIT' related to unit operations in the first description appears to be similar to button 'unit' in the second description, although the cases of the names differ.
2. The power button ('HOLD') button in the first description and button 'ON/OFF' in the second description both control the power function, suggesting potential similarity.

Considering these observations, two elements appear to be similar:
- Button related to units (case insensitive name comparison).
- Power button (function comparison).

Therefore, there are 2 elements that are common to both descriptions based on their functionality and intended actions.

By combining these analyses, the three numbers [A, B, C] are:
**[3, 3, 2]**

Figure 8: LLM-F1 for evaluating panel description task. $2C/(A + B) = 0.67$ here.

> *"'md*
> *{ prediction }*
>
> *"'*
> *The result should be start with 'The answer is 1/0', with 1 indicating they are the same and 0 indicating they are different. No analysis should be output if the answer is 1, but the reason should be provided if and only if the answer is 0.*

Fig. 9 demonstrates an example of how the LLM determines whether the prediction and the ground truth describe the same element. LJS-Acc is calculated on the entire test set.

## C.3 LJS-Acc for goal-based planning

For goal-based planning, we let LLM judge whether the plan in ground truth and prediction complete the same target. The prompt is:

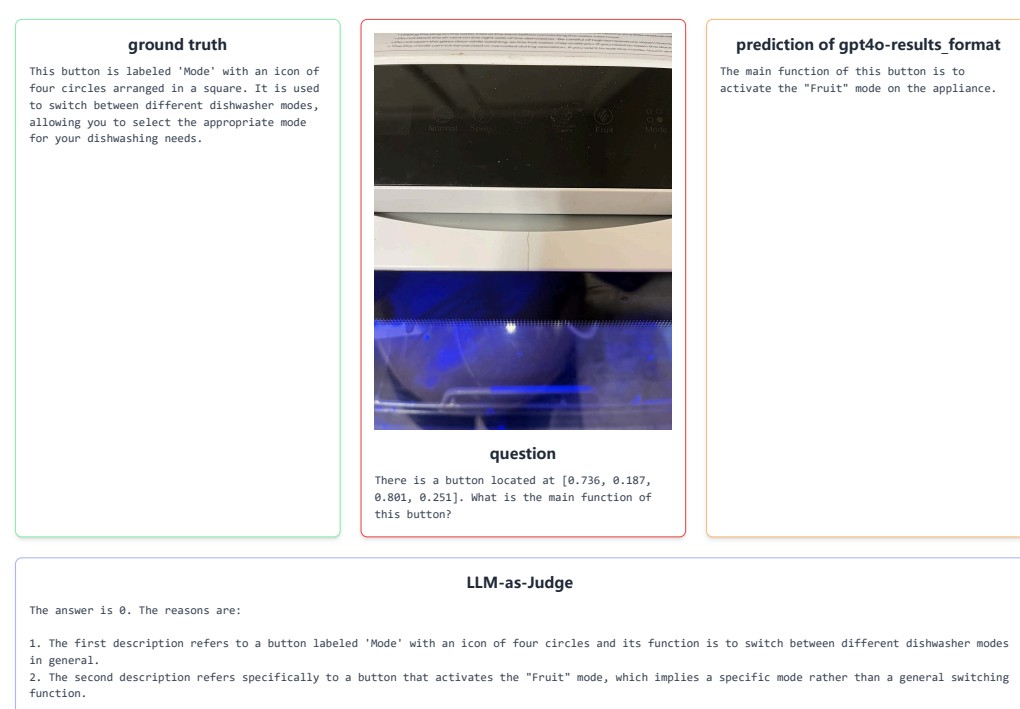

Figure 9: LLM-as-Judge in LLM-Acc for evaluating function estimation task. For the example, the judge result is false.

---

**The prompt for LJS-Acc in goal-based planning**

*The content in the following two markdowns describes the operational processes required to achieve a certain goal. The first one is the correct sequence of operational steps. Determine whether the second process is correct according to the first one. Some steps might be interchangeable, which should be consider correct. Overall, if there are no significant deviations (operating the wrong elements, omitting some key elements, etc.), it should be considered correct.*
*"'md*
*{ ground truth }*
*"'*

*"'md*
*{ prediction }*

*"'*
*The result should start with "The plan is 1/0" , where 1 indicates that the steps are correct, and 0 indicates they are not. No analysis should be output. The reason should be provided if and only if the plan is 0.*

---

Fig. 10 demonstrates an example on how to cool the room using an air conditioner.

## D Human evaluation on the LJS score

To evaluate the reliability of LJS in assessing the consistency between predicted answers and the ground truth (GT), we conducted a human evaluation study.

A total of 10 participants took part in the study. Each participant was presented with 10 randomly selected questions from each of the three tasks, with 7 model predictions per question, resulting in

To set the air conditioner to cool the room
quickly, follow these steps:
1. Press the '开关' button (labeled '开关') to
turn on the air conditioner. This will switch
on the air conditioner.
2. Press the '制冷' button (labeled '制冷') to
switch the air conditioner to cooling mode.
This will set the air conditioner to start
cooling the room.
3. Press the '温度' button with the arrow
pointing down icon to decrease the
temperature to the lowest comfortable
setting. Multiple presses may be required to
reach the desired temperature.
4. Optionally, if you want the air
conditioner to cool the room as quickly as
possible, you can press the '快速冷热' button
(labeled '快速冷热') to rapidly adjust the
cooling mode.

**prediction of claude-zs_format**

The plan is 0. The second process omits key steps such as turning on the air conditioner with the '开关' button and incorrectly includes steps like
setting the fan speed and adjusting the airflow direction, which are not mentioned in the first process.

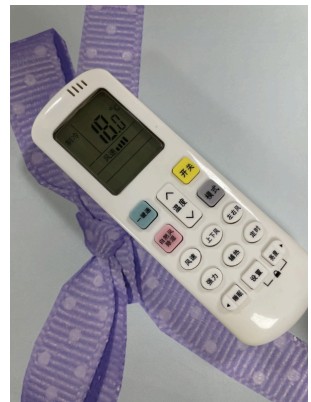

Figure 10: LLM-as-Judge in LLM-Acc for evaluating goal-based planning task. For the example, the judge result is false since the predicted plan omits the step to turn on the air conditioner.

Table 4: Human evaluation results on LJS(MAE).

| TASK | zero-shot | | fine-tuned | | | PPF | | avg. |
|---|---|---|---|---|---|---|---|---|
| | GPT-4o | Claude-3.7 | Yi-VL | LLAVA | Qwen-VL | *(w/ GPT-4o)* | *(w/ Claude)* | |
| description | 0.097 | 0.176 | 0.074 | 0.115 | 0.081 | 0.073 | 0.076 | 0.099 |
| function | 0.130 | 0.110 | 0.090 | 0.170 | 0.120 | 0.130 | 0.080 | 0.112 |
| planning | 0.090 | 0.140 | 0.110 | 0.120 | 0.120 | 0.150 | 0.110 | 0.120 |

$10 \times 3 \times 7 = 210$ Question-Answer-GT (QAG) triplets. The human annotators independently judged whether each prediction was semantically consistent with the corresponding GT.

The results are presented in Tab. 4. The MAE measures the average deviation between the LLM's consistency judgments and the majority vote of human annotators:

$$\text{MAE} = \frac{1}{N} \sum_{i=1}^{N} |S_{llm}^{(i)} - S_{human}^{(i)}|, \qquad (8)$$

where $S_{llm}^{(i)}$ and $S_{human}^{(i)}$ indicate the scores for each triplets judged by LLM and human, respectively. This metric provides an interpretable measure of how closely the LLM's evaluations align with human consensus. The average errors in the last column indicate that the LLM-as-judge makes mistakes at a rate of approximately 0.1, which is relatively stable across evaluations. This suggests that LLM-based metrics such as LLM-F1 and LLM-Acc can be considered acceptable, provided that the error margin is taken into account.

To examine whether subjective judgments from different participants are consistent for the same evaluation task, we further conducted a human study using $10 \times 3 = 210$ QAG triplets, covering three subtasks: *description*, *function*, and *planning*. Each of the ten participants independently scored all triplets. All participants produced nearly identical responses when presented with the same QAG triplet. To quantify inter-participant consistency, we organized all responses into a matrix $A = [a_{ij}]^{10 \times 70}$, where $a_{ij}$ denotes the judgment given by the $i$-th participant for the $j$-th triplet. Two metrics were computed to measure the degree of agreement:

Table 5: Consistency of human judgments across 10 participants.

| Metric | Description | Function | Planning |
|---|---|---|---|
| Average standard deviation | 0.0134 | 0.0043 | 0.0171 |
| Inconsistency rate | 0.0657 | 0.0014 | 0.0057 |

- **Average standard deviation** represents the mean of standard deviations over all triplets within a given task:

$$\text{average standard deviation} = \frac{1}{70}\sum_{j=1}^{70}\left[\sqrt{\frac{1}{10}\sum_{i=1}^{10}\left(a_{ij} - \frac{1}{10}\sum_{k=1}^{10}a_{kj}\right)^2}\right].$$

  This metric reflects the dispersion of participant scores for each triplet.

- **Inconsistency rate** measures the proportion of individual judgments that differ from the mode of responses to the same triplet:

$$\text{inconsistency rate} = \frac{1}{10\times 70}\sum_{j=1}^{70}\sum_{i=1}^{10}\mathbb{I}(a_{ij}\neq \text{Mo}_j(a_{ij})),$$

  where $\text{Mo}_j(a_{ij})$ denotes the mode of responses for the $j$-th triplet, and $\mathbb{I}(\cdot)$ is the indicator function.

The computed results are summarized in Table 5. The small average standard deviations and low inconsistency rates across all three subtasks indicate that the subjective judgments among different participants are highly consistent, demonstrating that participants exhibit a high degree of agreement, suggesting that the subjective human judgments in our evaluation can be considered consistent and reliable.

# E   Edge-side model

In PPF, a small VLM is deployed at the edge side to estimate the function of each detected elements by YOLO.

## E.1   Prompt & Visualization

For this model, the prompt is fixed and follows the formulation of

---

**The prompt for estimating function in the edge side**

*This image captures a panel of an appliance in real world. I will give you the type and location information of every module on this panel, you should enrich every module information by appending its description and function. Retain the original type and information details, following the original order and section format.*

*A {Type} from (left$_1$, top$_1$) to (right$_1$, bottom$_1$).*
*A {Type} from (left$_2$, top$_2$) to (right$_2$, bottom$_2$).*
*A {Type} from (left$_3$, top$_3$) to (right$_3$, bottom$_3$).*
*...*

---

Some examples are presented in Fig. 11 and Fig. 12. In the top row, we demonstrate the question-answer pairs for training the edge-side parsing model, while the bottom row presents two examples of enhanced results. Since all information in the edge side is organized via natural language and understandable to humans, users can also review the data to be sent before it is transmitted to the cloud, to ensure that there is no private information included in the message. The goal of PPF is to ensure that the private data hidden in images will not be utilized by the LLM provider.

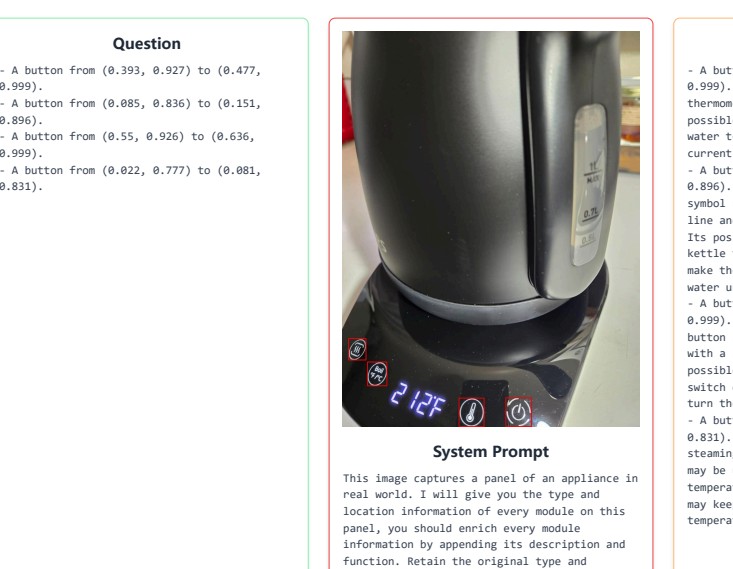

**Question**

- A button from (0.393, 0.927) to (0.477, 0.999).
- A button from (0.085, 0.836) to (0.151, 0.896).
- A button from (0.55, 0.926) to (0.636, 0.999).
- A button from (0.022, 0.777) to (0.081, 0.831).

**System Prompt**

This image captures a panel of an appliance in real world. I will give you the type and location information of every module on this panel, you should enrich every module information by appending its description and function. Retain the original type and information details, following the original order and section format.

**Prediction**

- A button from (0.393, 0.927) to (0.477, 0.999). It has '' on it and looks like A thermometer icon. This is a button. Its possible function is to control the viewing of water temperature. After pressing it, the current water temperature can be displayed. .
- A button from (0.085, 0.836) to (0.151, 0.896). It has 'Boil' on it and looks like A symbol representing boiling water with a wavy line and a drop of water.. This is a button. Its possible function is to start the electric kettle to boil water. When pressed, it can make the electric kettle start heating the water until it boils. .
- A button from (0.55, 0.926) to (0.636, 0.999). It has '' on it and looks like A power button icon, typically represented by a circle with a line at the top.. This is a button. Its possible function is to control the power switch of the electric kettle. Pressing it can turn the electric kettle on or off. .
- A button from (0.022, 0.777) to (0.081, 0.831). It has '' on it and looks like A steaming kettle icon. This is a button which may be used to set or activate the constant temperature function. After being pressed, it may keep the electric kettle at a constant temperature. .

Figure 11: An example of parser with YOLO results as the question.

**Detector Result**          **VLM Enhancement**

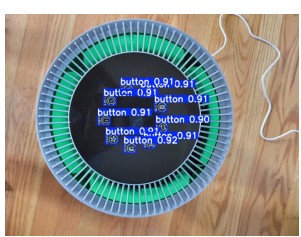

📍 *A knob from (0.348, 0.643) to (0.679, 0.858).*

👀 *It has '**START + 30 SEC TIME WEIGHT PROG LEVEL**' on it .*

⚙ *This is a knob that may be used to select different functions of the microwave oven, such as starting heating, increasing by 30 seconds each time, defrosting, setting the time, defrosting by weight, or adjusting the power level. Rotating this knob can select different options.*

📍 *button from (0.341, 0.542) to (0.384, 0.585).*

👀 *It has '**SPEED**' on it and looks like **fan** icon.*

⚙ *Its possible function is to adjust the wind speed of the air purifier. After pressing it, the wind speed can be adjusted.*

Figure 12: Examples of the parsing of button and knob that are enhanced on the edge-side model.

## E.2 Latency Analysis

As for the deployment of the edge-side model, we refer to [45, 2, 29, 57] and test the latency of the edge-side parser. The edge-side hardware is a realmeNeo 7 running Android 15, equipped with a Dimensity 9300 processor, 16GB RAM, and 1TB ROM, and deploying VLM using Ollama.

Using the input panel image with a resolution of $1536 \times 768$, which is tokenized into 1540 tokens in QwenVL, we test the FPS of YOLO on our device and achieve 11.9 (frame/s), which equals 18326 (token/s). Besides, we obtain the speed of the deployed VLM on the edge side as in Tab. 6.

However, we infer that the performance of edge-side models might still have untapped performance potential. The output speed of the edge-side model in [45] can be greater than 30 token/s, nearly

Table 6: The speed of the deployed VLM

| Model | Input Speed | Output Speed | Storage | Memory |
|---|---|---|---|---|
| Qwen2.5VL-3B | 1628.58 (token/s) | 6.68 (token/s) | 3.2 (GB) | 3.7 (GB) |

5 times faster than our implementation. This suggests promising room for further optimization in future work. Overall, our implementation indicate that it is possible to deploy a 3B VLM combined with YOLO on an ordinary Android phone. We hope this could have addressed your convern.

## F   Visualization comparison of model predictions

From Fig. 13 to Fig. 16, we compare the zero-shot and fine-tuned results of Qwen-2.5-7B VL model on four tasks, which demonstrate that zero-shot model is short of knowledge on panel understanding and operation.

**Panel Description**

- *Button 'Set Temperature' - This button displays the current temperature setting on the air conditioner. You can also use the up and down arrows to adjust the temperature.*
- Button 'Mode' - This button allows you to select the operating mode of the air conditioner. Common modes include:
  -Auto: The air conditioner automatically selects the most suitable mode (cooling, heating, or drying) based on the temperature settings and room conditions.
  -Cool: The air conditioner cools the room.
  -Dry: The air conditioner removes moisture from the air.
  -Heat: The air conditioner warms the room.
- Button 'On/Off' - This button turns the air conditioner on or off.
- Button 'Fan Speed' - This button allows you to adjust the fan speed of the air conditioner. Higher speeds can provide faster cooling or heating, while lower speeds can be more energy-efficient.
- Button 'Temp Up' - This button increases the temperature setting on the air conditioner.
- Button 'Temp Down' - This button decreases the temperature setting on the air conditioner.
- Button 'Swing' - This button controls the oscillation of the air conditioner's fan or blades. Oscillation can help distribute the airflow more evenly throughout the room.
- Button 'Sleep' - This button sets the air conditioner to a \"sleep\" mode, which gradually decreases the temperature over a set period of time to ensure you fall asleep comfortably.
- Button 'Timer On' - This button sets the timer to turn the air conditioner on at a specific time.
- Button 'Timer Off' - This button cancels any previously set timer and stops the air conditioner from turning on at a scheduled time.
- Button 'Reset Lock' - This button resets the lock function, which can be useful if you've accidentally locked the remote.
- Button 'LED Display' - This button controls the display settings of the remote control, such as turning the display on or off.
- Button 'Turbo' - This button increases the cooling or heating power temporarily, providing a quick boost to the air conditioner's performance.
- Button 'Air Direction' - This button adjusts the direction of the airflow from the air conditioner. It allows you to direct the airflow to a specific area of the room.

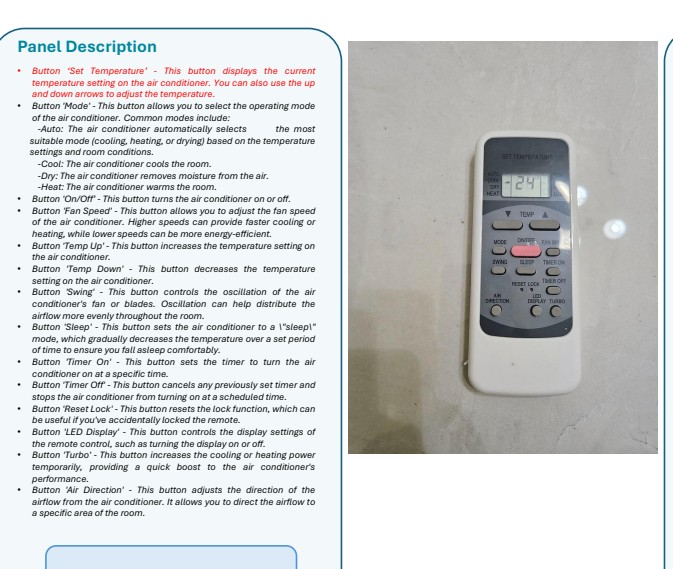

Qwen2.5VL-zero-shot

**Panel Description**

- Button 'AIR DIRECTION' - This button adjusts the direction of the air flow. Pressing it allows you to change the air outlet direction to suit your comfort.
- Button 'TIMER ON' - This button sets the air conditioner timer to start the air conditioner automatically. Pressing it enables the timer mode for scheduled operation.
- Button 'FAN SPEED' - This button adjusts the fan speed of the air conditioner. Pressing it allows you to switch between different fan speed settings.
- Button 'SLEEP' - This button activates the sleep mode of the air conditioner. Pressing it adjusts the air conditioner's operation to create a comfortable environment for sleeping.
- Button 'TIMER OFF' - This button sets the air conditioner timer to turn off the air conditioner automatically. Pressing it enables the timer to switch off the unit after a specified period.
- Button 'TEMP' (Up arrow) - This button increases the temperature of the air conditioner. Pressing it raises the temperature setting.
- Button 'MODE' - This button switches the operating mode of the air conditioner. Pressing it allows you to change between modes such as cooling, heating, dehumidifying, and air supply.
- Button 'LED DISPLAY' - This button controls the LED display settings of the air conditioner. Pressing it can turn the display on or off, or switch the display modes.
- Button 'TEMP' (Down arrow) - This button lowers the set temperature of the air conditioner. Pressing it decreases the temperature setting.
- Button 'SWING' - This button controls the swing function of the air conditioner. Pressing it makes the air outlet swing, either up and down or left and right, to distribute air more evenly.
- Button 'ON/OFF' - This button turns the air conditioner on or off. Pressing it powers the air conditioner up or down.
- Button 'RESET LOCK' - This button resets the air conditioner settings or lifts the lock. Pressing it restores the default settings or unlocks certain functions.
- Button 'TURBO' - This button activates the turbo mode of the air conditioner. Pressing it increases the cooling or heating efficiency to provide a rapid temperature adjustment.

Qwen2.5VL-finetune

**Panel Description**

- Button 'TEMP (with upward arrow symbol)' - This button is used to increase the set temperature.
- Button 'TEMP (with downward arrow symbol)' - This button is used to decrease the set temperature.
- Button 'MODE' - This button is used to switch between different modes (AUTO, COOL, DRY, HEAT).
- Button 'ON/OFF' - This button is used to power the air conditioner on or off.
- Button 'FAN SPEED' - This button is used to adjust the fan speed.
- Button 'SWING' - This button is used to control the swing function of the air direction.
- Button 'SLEEP' - This button is used to activate or deactivate the sleep mode.
- Button 'TIMER ON' - This button is used to set the timer to turn the air conditioner on at a specified time.
- Button 'TIMER OFF' - This button is used to set the timer to turn the air conditioner off at a specified time.
- *Button 'RESET (with lock symbol)' - This button is used to reset the settings to their default values or to lock the current settings.*
- Button 'AIR DIRECTION' - This button is used to adjust the direction of the air flow.
- Button 'LED DISPLAY' - This button is used to control the LED display.
- Button 'TURBO' - This button is used to activate the turbo mode for faster cooling or heating.

GPT-4o

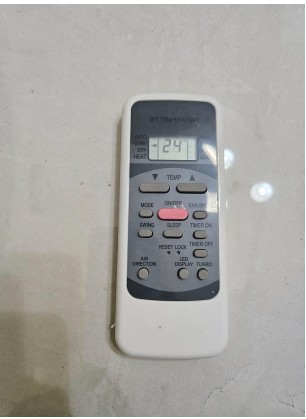

**Panel Description**

- Button 'TEMP' () - This button adjusts the air-conditioning temperature. Pressing it allows relevant operations on the temperature setting.
- Button 'ON/OFF' () - This button controls turning the air conditioner on or off. Pressing it executes this operation.
- Button 'AIR DIRECTION' () - This button adjusts the air outlet direction of the air conditioner. Pressing it changes the outlet angle.
- Button 'TEMP' () - This button adjusts the air-conditioning temperature. Pressing it allows adjustment of the set temperature.
- Button 'LED DISPLAY' () - This button controls the LED display screen of the air conditioner. Pressing it allows operations such as turning the display on/off or switching modes.
- Button 'TIMER ON' () - This button turns on the air conditioner timer. Pressing it allows setting a scheduled run time.
- Button 'TIMER OFF' () - This button sets the air conditioner to turn off at a scheduled time. Pressing it allows setting the turn-off time.
- Button 'FAN SPEED' () - This button adjusts the air conditioner fan speed. Pressing it switches between different speed settings.
- Button 'SLEEP' () - This button turns on the sleep mode of the air conditioner. Pressing it activates sleep mode.
- Button 'TURBO' () - This button turns on the strong mode of the air conditioner. Pressing it operates the air conditioner at higher power.
- Button 'MODE' () - This button switches the air conditioner mode. Pressing it cycles through different modes such as cooling, heating, dehumidifying, air supply, etc.
- Button 'SWING' () - This button controls the air swing function of the air conditioner. Pressing it starts the air direction swing up and down or left and right.

PPF(ours)

Figure 13: An example demonstrating models' ability in the *panel description* task. The zero-shot VLM mistakenly identifies the character "set temperature" at the top as a button. On the other hand, GPT-4o misidentifies the reset indicator as a button.

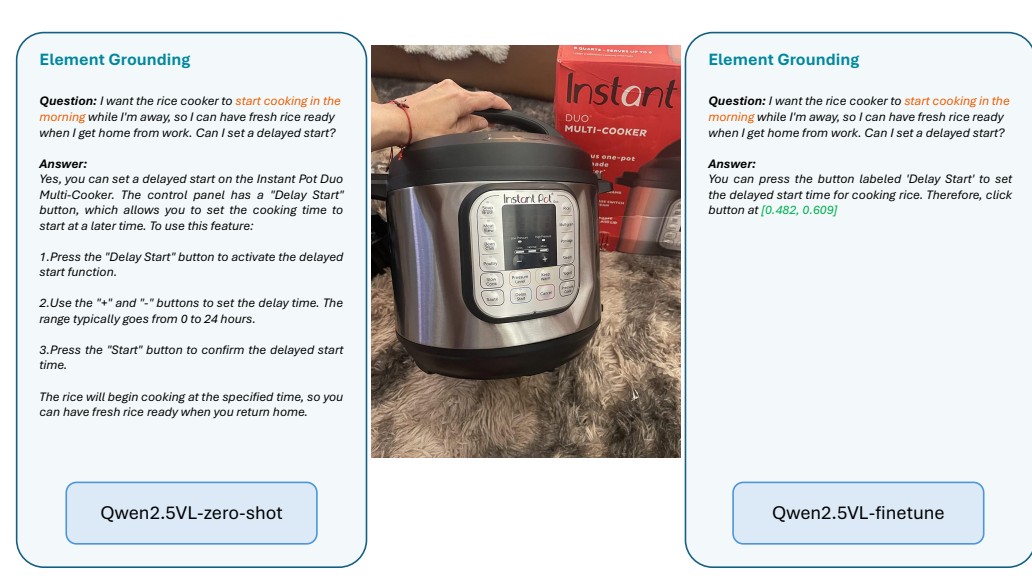

## Element Grounding

**Question:** *I want the rice cooker to start cooking in the morning while I'm away, so I can have fresh rice ready when I get home from work. Can I set a delayed start?*

**Answer:**
*Yes, you can set a delayed start on the Instant Pot Duo Multi-Cooker. The control panel has a "Delay Start" button, which allows you to set the cooking time to start at a later time. To use this feature:*

*1.Press the "Delay Start" button to activate the delayed start function.*

*2.Use the "+" and "-" buttons to set the delay time. The range typically goes from 0 to 24 hours.*

*3.Press the "Start" button to confirm the delayed start time.*

*The rice will begin cooking at the specified time, so you can have fresh rice ready when you return home.*

Qwen2.5VL-zero-shot

## Element Grounding

**Question:** *I want the rice cooker to start cooking in the morning while I'm away, so I can have fresh rice ready when I get home from work. Can I set a delayed start?*

**Answer:**
*You can press the button labeled 'Delay Start' to set the delayed start time for cooking rice. Therefore, click button at [0.482, 0.609]*

Qwen2.5VL-finetune

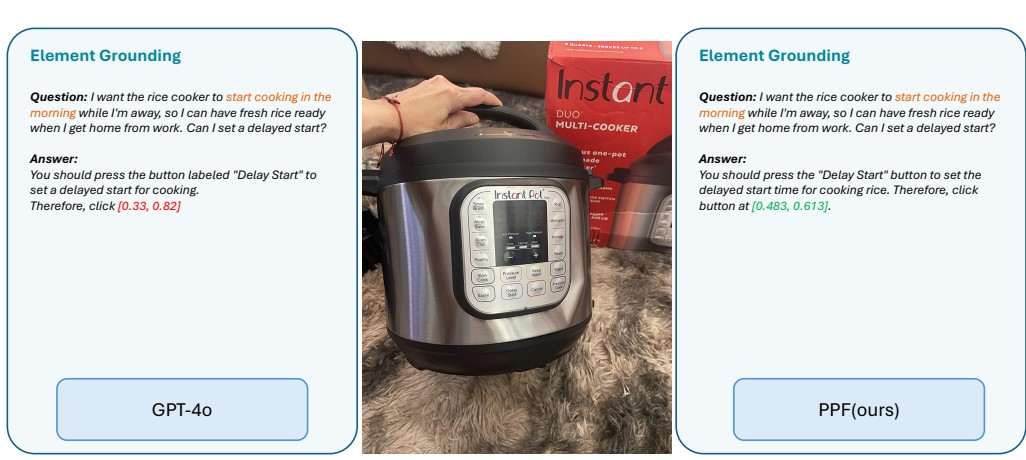

## Element Grounding

**Question:** *I want the rice cooker to start cooking in the morning while I'm away, so I can have fresh rice ready when I get home from work. Can I set a delayed start?*

**Answer:**
*You should press the button labeled "Delay Start" to set a delayed start for cooking. Therefore, click [0.33, 0.82]*

GPT-4o

## Element Grounding

**Question:** *I want the rice cooker to start cooking in the morning while I'm away, so I can have fresh rice ready when I get home from work. Can I set a delayed start?*

**Answer:**
*You should press the "Delay Start" button to set the delayed start time for cooking rice. Therefore, click button at [0.483, 0.613].*

PPF(ours)

Figure 14: An example demonstrating models' ability in the *element grounding* task. The zero-shot VLM cannot generate a coordinate for visualization, while GPT-4o has produced an incorrect 2d coordinate.

**Function Inference**

*Question:* There is a button located at [0.79, 0.625, 0.831, 0.655]. Can you explain the dual functionality of this button?

*Answer:* The button located at [0.79, 0.625, 0.831, 0.655] on the microwave appears to be the "Start" button. In most microwaves, the "Start" button is used to initiate the cooking process. It has a dual functionality: it can start the microwave for a set time (when you press it once) or it can start the microwave immediately (when you press it twice in quick succession). The exact behavior can vary depending on the model and user manual. It's important to refer to the user manual for the specific microwave for precise instructions.

Qwen2.5VL-zero-shot

**Function Inference**

*Question:* There is a button located at [0.79, 0.625, 0.831, 0.655]. Can you explain the dual functionality of this button?

*Answer:* This button has a two-in-one function labeled 'START/+30Sec./0'. It can start the microwave oven, and pressing it twice will increase the cooking time by 30 seconds. Additionally, pressing it three times will reset the time to 0.

Qwen2.5VL-finetune

**Function Inference**

*Question:* There is a button located at [0.79, 0.625, 0.831, 0.655]. Can you explain the dual functionality of this button?

*Answer:*
The button is labeled "STOP (Cancel)" and serves two functions: it stops the microwave during operation and cancels any current settings or operations.

GPT-4o

**Function Inference**

*Question:* There is a button located at [0.79, 0.625, 0.831, 0.655]. Can you explain the dual functionality of this button?

*Answer:*
This button located at [0.789, 0.625, 0.831, 0.656] has dual functionality:

1. **START**: When pressed, it starts the microwave oven's operation.

2. **+30Sec.**: Additionally, it can set a 30-second heating time, starting the microwave for that duration.

PPF(ours)

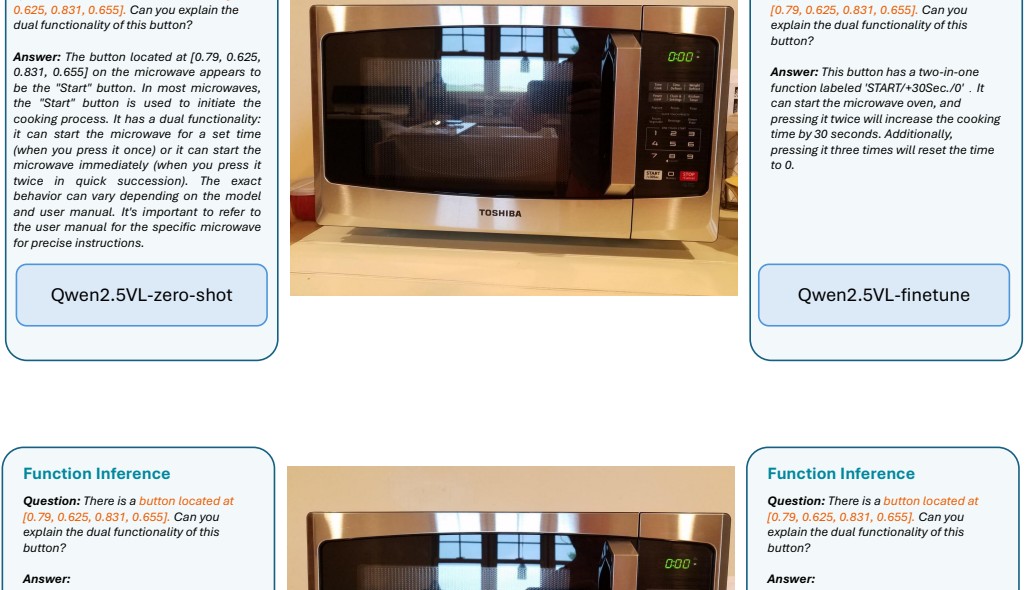

Figure 15: An example demonstrating models' ability in the *function inference* task. The zero-shot model failed to recognize the function of "+30 sec", while the result of GPT-4o is completely wrong, which is entirely different from the specified button.

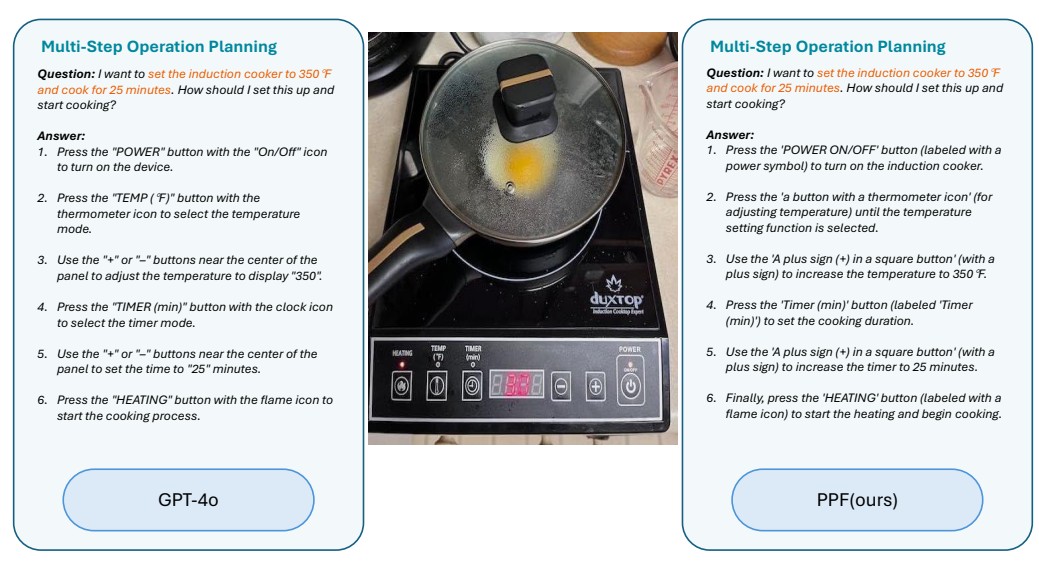

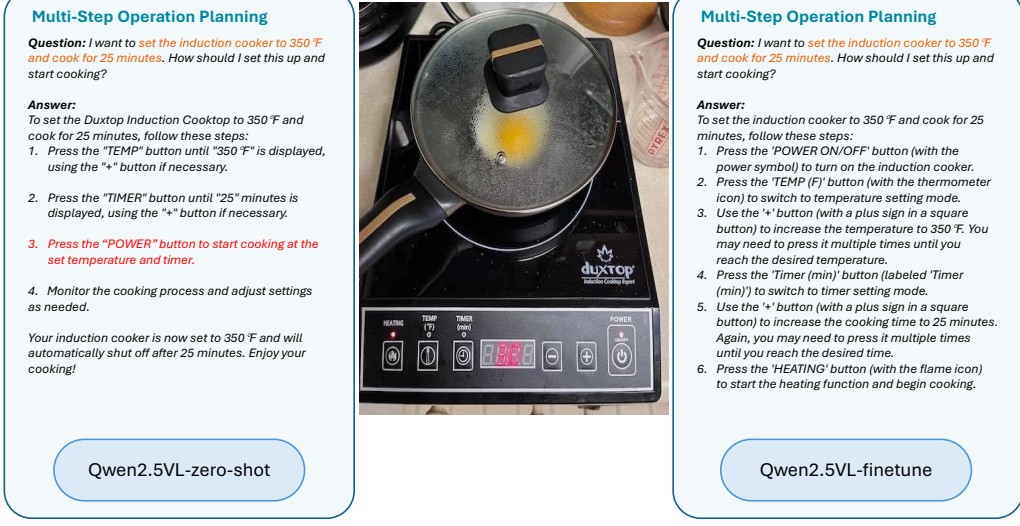

Figure 16: An example demonstrating models' ability in the multi-step operation planning task. The zero-shot one advises to press the "power" button at the third step, which will turn off the induction cooker. This action deviates from the planning goal.

