# OpenReview forum: "PUO-Bench: A Panel Understanding and Operation Benchmark with A Privacy-Preserving Framework"
_NeurIPS.cc/2025/Datasets_and_Benchmarks_Track — NeurIPS 2025 Datasets and Benchmarks Track poster_

### Official Review · Reviewer_X5m1 · 2025-06-04

**Rating:** 5
**Confidence:** 3

**Summary:**

This paper introduces PUO-Bench, a large-scale benchmark for interpreting and operating physical control panels. The dataset contains 19K panel images spanning 11 appliance categories and 430K vision-language instruction pairs, annotated with element type, position, icon, text, and function. Four tasks are defined: Panel Description, Element Grounding, Functional Inference, and Multi-Step Operation Planning. The authors evaluate zero-shot and fine-tuned VLMs, demonstrating that fine-tuning on PUO substantially improves performance. They also propose a Privacy-Preserving Framework (PPF) that runs lightweight object detection and description on edge devices, sending only structured, non-sensitive data to cloud-based LLMs for reasoning. Experiments show PPF achieves near fine-tuned VLM accuracy while avoiding raw image uploads.

**Additional Feedback:**

N/A

**Dataset Code Accessibility:**

Yes

**Dataset Code Comments:**

The dataset can be easily accessed

**Ethical Considerations:**

No, there are no or only very minor ethics concerns

**Final Justification:**

The author's rebuttal has alleviated my concerns, so I raised the rating to 5. Why not a higher rating: lack of regard for panels under extreme or challenging conditions

**Limitations Weaknesses:**

Limited Domain Coverage: PUO focuses on household appliances; industrial, medical, or commercial panels remain unaddressed. Extreme conditions like severe occlusion or poor lighting are not quantified, potentially limiting robustness.

Incomplete Edge/Latency Analysis: PPF’s feasibility on resource-constrained devices is unclear—no latency measurements or minimum hardware requirements are provided. Real-time performance and energy costs are not evaluated.

Privacy Assumptions Underexplored: Although only structured data is sent, attackers might still infer sensitive details. Communication security (e.g., encryption) and potential inference attacks are not analyzed.

Generalization to Unseen Panels Untested: There is no dedicated experiment on zero-shot generalization to novel device types or unseen element layouts, leaving open the question of model robustness in truly open settings.

Deployment and Ethical Considerations: The paper lacks discussion of integration costs (hardware, software) and potential safety risks in critical applications (e.g., medical devices), as well as broader ethical or regulatory implications.

**Strengths Contributions:**

Novel, Real-World Dataset: The PUO dataset fills a gap by focusing on real physical panels (e.g., microwaves, ovens, air conditioners), with detailed annotations for type, location, and function of each control element.

Comprehensive Task Suite: By defining four distinct tasks—description, grounding, inference, and multi-step planning—the benchmark evaluates a wide spectrum of panel understanding and instruction-following abilities.

Privacy-Preserving Design: PPF’s two-stage approach (edge-side detection and description + cloud-side reasoning) mitigates privacy risks without notably sacrificing accuracy.

Thorough Evaluation: The paper benchmarks multiple state-of-the-art VLMs (Yi-VL, LLAVA-NeXT, Qwen-VL) in zero-shot and fine-tuned settings, compares different cloud LLMs, and ablates PPF components, offering clear insights into performance trade-offs.

Clear Presentation: The writing is well-structured; figures and tables concisely illustrate data distribution, sample annotations, and quantitative results.

---

> ### Author Rebuttal · Authors · 2025-07-31
>
> We are genuinely grateful for your detailed assessment of our work and the valuable insights you have shared. Your constructive feedback has significantly contributed to the refinement and advancement of our study.
>
> **1. [Re Weakness 1: Domain Coverage.]**
>
> Thanks for the helpful comments. We explain as follows:
>
> - ###### Scope of PUO Benchmark
>
>   - Since ours is the first work in this domain, PUO currently focuses on the most common types of panel holders—**household appliances**, as illustrated in Fig. 3(a). We chose this category because it is widely accessible and representative of real-world panel operation scenarios.
>   - That said, we plan to **broaden the scope of PUO to professional domains** in future work, including medical devices, industrial equipment, electrical measurement instruments, control panels in transportation systems, etc.
>   - However, these domains often require domain-specific expertise for data collection and annotation. As such, we need more time and resources to build high-quality datasets in these areas. Fortunately, the methodology and pipeline we established for constructing the current PUO benchmark can serve as a solid reference for extending to these professional contexts.
> - ###### Panels in Extreme Conditions
>
>   - Regarding panels under extreme or challenging conditions (e.g., occlusion, poor lighting), we **did consider these scenarios** during the benchmark design phase. However, we ultimately chose to **exclude images with severe quality issues**, as even human annotators struggle to determine the functions of operable elements in such cases.
>   - We believe that excluding these low-quality samples is reasonable, as our goal is to create a meaningful benchmark that reflects practical and evaluative system conditions. Notably, in several failure cases (with scores close to zero on our benchmark), we observed that both end-to-end VLMs and our proposed PPF method frequently produce **recognition errors and hallucinations** when the visual clarity of the panel is compromised.
>   - These findings further support our decision to filter out such data for now, while leaving open the possibility of addressing extreme-condition robustness in future versions of PUO.
>
> **2. [Re Weakness 2: Edge/Latency Analysis.]**
>
> - As for the deployment of the edge-side model, we refer to [C1, C2, C3, C4] and test the latency of the edge-side parser. The edge-side hardware is a realmeNeo 7 running Android 15, equipped with a Dimensity 9300 processor, 16GB RAM, and 1TB ROM, and deploying VLM using Ollama.
>
>   1. Using the input panel image with a resolution of 1536×768, which is tokenized into 1540 tokens in QwenVL, we test the FPS of YOLO on our device and achieve 11.9 (frame/s), which equals 18326 (token/s).
>   2. Besides, we obtain the speed of the deployed VLM on the edge side as follows:
>
> |Model|Input Speed (token/s)|Output Speed (token/s)|Storage (GB)|Memory (GB)|
> | :------------: | :---------------------: | :----------------------: | :------------: | :-----------: |
> |Qwen2.5VL-3B|1628.58|6.68|3.2|3.7|
>
> - However, we infer that the performance of edge-side models might still have untapped performance potential. The output speed of the edge-side model in [C1] can be greater than 30 token/s, nearly 5 times faster than our implementation. This suggests promising room for further optimization in future work.
> - Overall, our implementation indicate that it is possible to deploy a 3B VLM combined with YOLO on an ordinary Android phone. We hope this could have addressed your convern.
>
> [C1] X.Lu, et al, "Bluelm-v-3b: Algorithm and system co-design for multimodal large language models on mobile devices," CVPR 2025.
>
> [C2] M.Abdin, et al, "Phi-4 technical report," arXiv, 2024.
>
> [C3] P.R. Guerrero, et al, "Efficient Deployment of Vision-Language Models on Mobile Devices: A Case Study on OnePlus 13R," arXiv, 2025.
>
> [C4] J.Xiao, et al. "Understanding Large Language Models in Your Pockets: Performance Study on COTS Mobile Devices," arXiv, 2024.
>
> **3. [Re Weakness 3: Privacy Assumptions.]**
>
> We understand your concern. We address this from the following aspects:
>
> - Here we highlight that the specific content of structured data is strictly defined within the SFT's edge-side VLM. Please refer to the supplementary material's Section.D, Fig.11, and Fig.12. Each extracted element will be represented in the following format: `A {type} from (x1, y1) to (x2, y2). It has '{text}' on it and looks like '{icon}'. {function}`. This strictly defined format effectively prevents privacy leaks.
> - Additionally, since this data is organized via natural language and understandable to humans, users can also review the data to be sent before it is transmitted to the cloud, to ensure that there is no private information included in the message.
> - As for encryption, this is commonly guaranteed by networking protocols. The goal of PPF is to ensure that the private data hidden in images will not be utilized by the LLM provider.
>
> **4. [Re Weakness 4: Generalization to Unseen Panels.]**
>
> - Thank you for the comments. We have resplit all samples to ensure that the training set and test set do not overlap, and then compared the results between zero-shot Qwen and fine-tuned Qwen. The results are demonstrated in the table:
>
> |   Method   | desc ($C_{\text{CIDEr}}$) | desc ($C_{\text{BERT-S}}$) | desc (LJS-F1) | Grd(Coord) | Grd (Acc) | Func (BERT-S) | Func (LJS-Acc) | Planning (LJS-Acc) |
> | :----------: | :-------: | :-------: | :-------------: | :----------: | :---------: | :-------------: | :--------------: | :------------------: |
> | Qwen (zs) |  0.82  |  0.840  |     0.395     |    0.0    |    0.0    |     0.876     |     0.235     |       0.116       |
> | Qwen (ft) |  18.46  |  0.907  |     0.445     |   0.996   |   0.528   |     0.913     |     0.267     |       0.226       |
>
> The results show that the model fine-tuned on a cross-category setting still performs better than the one without fine-tuning, proving that our dataset also supports unseen panels testing and can make it work better.
>
> - The results show that the model fine-tuned on a cross-category setting still performs better than the one without fine-tuning, proving that our dataset also supports unseen panels testing and can make it work better.
>
> **5. [Re Weakness 5: Deployment and Ethical Considerations.]**
>
> Thanks for the comments, we will add relevant discussion in our new version:
>
> - **Hardware and software:**  supervised fine-tuning details can be found in Section.A in the supplemental material. As for the deployment of edge-side parser, please refer to the response to `Q1: Incomplete Edge/Latency Analysis`. We will also add this new information to our new version.
> - **Safety risks:**  While our PUO benchmark and PPF aim to advance human-machine interaction in real-world environments, we acknowledge several safety and ethical implications. First, incorrect panel interpretation could lead to unsafe physical actions (e.g., unintended device activation). We emphasize that autonomous operation should only occur under strict safety protocols and human oversight, especially for high-risk appliances. Second, the model may exhibit biases due to dataset limitations, and adversarial vulnerabilities remain a concern in real-world conditions. Future work should incorporate robustness testing and fairness audits. Finally, we caution against misuse in unauthorized access scenarios and advocate for responsible deployment guided by AI ethics principles.
> - **Ethical and regulatory implications:**  The panel images in our dataset are collected from publicly available website. All images were originally uploaded by users voluntarily and do not involve direct data collection from individuals. While the data are publicly accessible, we acknowledge potential incidental exposure of private contexts; thus, our Privacy-Preserving Framework (PPF) further ensures no raw images are transmitted in deployment. The system is designed for human-in-the-loop assistance, not autonomous control, aligning with safety and accountability principles in emerging AI regulations. We advocate responsible use in authorized, non-invasive applications.

---

> > ### Comment · Reviewer_X5m1 · 2025-08-04
> >
> > The author's rebuttal has alleviated my concerns, so I raised the rating to 5. Why not a higher rating: lack of regard for panels under extreme or challenging conditions

---

> > > ### Author Response · Authors · 2025-08-05
> > > **Official Comment by Authors**
> > >
> > > Dear Reviewer X5m1，
> > >
> > > We sincerely appreciate the reviewer for acknowledging our work and considering raising the score. Your constructive suggestions really help improve our manuscript.
> > >
> > > Best regards,
> > >
> > > Authors of #1551

---

> > ### Author Response · Authors · 2025-08-06
> > **Discussion on extreme condition**
> >
> > Thanks for your feedback! Regarding panels under extreme conditions, we provide the following explanations to address your concern. We will also include this analysis in the final version.
> >
> > ***From the perspective of data collection***
> >
> > - Image quality is a crucial factor we had to consider at the beginning of benchmark establishment. However, annotators cannot recognize functions on panels in poor-quality images. Thus, we use a recent image quality assessment model, QAlign[C5], to score each image, and remove those with scores lower than 3.5 (the maximum score is 5), while others (score \> 3.5) are maintained in our benchmark.
> > - However, we still identified some images that are not easy to recognize and were captured in challenging conditions. Here are some representative examples:
> >
> >   - Perspective problems: `tb_Induction_Cooker_unknown_5c34165d.jpg`, `tb_Microwave_Oven_Panasonic_fa342d93.jpg`, `jd_Air_Purifier_IAM_de28262d.jpg`
> >
> >   - Blur panels: `tb_Microwave_Oven_Konka_13316dc8.jpg`, `AC_Remote_Amazon_03566.jpg`
> >
> >   - Dark environments: `tb_Microwave_Oven_Panasonic_fa342d93.jpg`, `Induction_Cooker_iSiLER_1800W_05578.jpg`, `Oven_Breville_BOV800XL_07641.jpg`
> >
> >   - Panels far from the camera: `Dryer_COSTWAY_Compact_04716.jpg`, `Induction_Cooker_Duxtop_P961LS_05479.jpg`, `jd_Air_Purifier_IAM_de28262d.jpg`
> >
> >   - Electronic touch screens/panels: `jd_Air_Purifier_A.O.SMITH_d495b806.jpg`, `tb_Dishwasher_Haier_5ab899f3.jpg`, `jd_Washing_Machine_Casarte_1ac969db.jpg`
> > - Since all images are collected from the internet rather than captured in an experimental environment, our PUO dataset actually includes various scenes in which these panels exist.
> > - For annotation, we searched for images similar to these unclear images on "Google Images" and found clear versions to help annotate the collected unclear images.
> >
> > ***From the perspective of performance***
> >
> > - To show how VLM performs on these images in challenging conditions, we extracted 50 unclear images from the test dataset and compared them with the results of the entire test set:
> >
> > | Method | extra cond. | desc (LJS-F1) | Grd (Acc) | Func (LJS-Acc) | Planning (LJS-Acc) |
> > | :------: | :-------------: | :-------------: | :---------: | :--------------: | :------------------: |
> > | GPT-4o | ×           |     **0.545**     |   **0.063**   |     **0.297**     |       **0.294**       |
> > | GPT-4o | ✔            |     0.340     |    0.0    |     0.216     |       0.208       |
> > |  PPF  | ×            |     **0.706**     |   **0.673**   |     **0.630**     |       **0.420**       |
> > |  PPF  | ✔            |     0.475     |   0.458   |     0.342     |       0.250       |
> >
> > - The results show that challenging conditions indeed affect the performance of PUO methods. However, models fine-tuned on our training set perform better than GPT-4o.
> >
> > [C5] Wu. H, et al, "Q-align: Teaching LMMS for visual scoring via discrete text-defined levels," ICML, 2024.

---

### Official Review · Reviewer_BDQR · 2025-06-12

**Rating:** 5
**Confidence:** 4

**Summary:**

This paper introduces PUO-Bench, a benchmark for Panel Understanding and Operation, aiming to address the gap in real-world interactions with control panels. The benchmark includes over 19,000 annotated panel images and 430,000 instruction-following QA pairs.

Experimental results show that existing zero-shot VLMs perform poorly on panel understanding tasks, while models fine-tuned on the dataset achieve significantly better results.

Further, the authors propose a Privacy-Preserving Framework that performs panel understanding on edge devices and only transmits essential information to the cloud for complex reasoning, reducing privacy risks by avoiding raw image transmission.

In Summary, this work provides valuable resources and methods to advance applications in human-computer interaction and robotics.

**Dataset Code Accessibility:**

Yes

**Dataset Code Comments:**

Datast is released on Huggingface. However, a detailed readme and usage guidance is recommend.

**Ethical Comments:**

This article mainly focuses on various remote controls and control board information. There might be some issues related to commercial brands and logos, but overall these are very minor issues and should not raise any ethical concerns.

**Ethical Considerations:**

No, there are no or only very minor ethics concerns

**Final Justification:**

The rebuttal addressed my issue. I will keep my score.

**Limitations Weaknesses:**

- In the privacy-related experiments, the on-device models are also around 3B in size, which makes it practically difficult to run on current smartphones and similar devices. Are there any experiments or case studies that support using smaller models for on-edge decision-making? Alternatively, are there any examples where techniques such as quantization have been applied to enable deployment of such models on common mobile devices?
- The publicly available dataset on Huggingface does not follow the common Huggingface format; it is simply a ZIP file, and the dataset card is not well written. We hope this can be improved.

**Strengths Contributions:**

* The paper introduces a large and diverse dataset of real-world control panels with rich instruction-following QA pairs. This is a novel topic that has not been explored before.
* To address privacy concerns, the authors propose a PPF framework, which holds practical value for real-world applications.
* The paper presents solid experimental comparisons between zero-shot and fine-tuned VLMs, clearly highlighting the limitations of current models and the advantages of task-specific tuning.
* The four benchmark tasks—panel description, element localization, function inference, and multi-step planning—cover key aspects of panel understanding and are well-designed to reflect real-world challenges.

---

> ### Author Rebuttal · Authors · 2025-07-31
>
> We sincerely thank you for your comprehensive examination of our paper and value the thoughtful feedback you have offered. Your helpful suggestions have played a crucial role in improving the overall quality of our research.
>
> **1. [Re Weakness 1: Deployment on Common Mobile Devices.]**
>
> We understand your concern. We address it as follows:
>
> - As for the deployment of the edge-side model, we refer to [C1, C2, C3, C4] and test the latency of the edge-side parser. The edge-side hardware is a realmeNeo 7 running Android 15, equipped with a Dimensity 9300 processor, 16GB RAM, and 1TB ROM, and depolying VLM using Ollama.
>
>   1. Using the input panel image with a resolution of 1536×768, which is tokenized into 1540 tokens in QwenVL, we test the FPS of YOLO on our device and achieve 11.9 (frame/s), which equals 18326 (token/s).
>   2. Besides, we obtain the speed of the deployed VLM on the edge side as follows:
>
> |Model|Input Speed (token/s)|Output Speed (token/s)|Storage (GB)|Memory (GB)|
> | :------------: | :---------------------: | :----------------------: | :------------: | :-----------: |
> |Qwen2.5VL-3B|1628.58|6.68|3.2|3.7|
>
> - However, we infer that the performance of edge-side models might still have untapped performance potential. The output speed of the edge-side model in [C1] can be greater than 30 token/s, nearly 5 times faster than our implementation. This suggests promising room for further optimization in future work.
> - Overall, our implementation and previous cases indicate that it is possible to deploy a 3B model on the edge side. We hope this has addressed your concern.
>
> [C1] X.Lu, et al, "Bluelm-v-3b: Algorithm and system co-design for multimodal large language models on mobile devices," CVPR 2025.
>
> [C2] M.Abdin, et al, "Phi-4 technical report," arXiv, 2024.
>
> [C3] P.R. Guerrero, et al, "Efficient Deployment of Vision-Language Models on Mobile Devices: A Case Study on OnePlus 13R," arXiv, 2025.
>
> [C4] J.Xiao, et al. "Understanding Large Language Models in Your Pockets: Performance Study on COTS Mobile Devices," arXiv, 2024.
>
> **2. [Re Weakness 2: Huggingface File Format.]**
>
> Thank you for your constructive feedback! We sincerely apologize for the inconvenience caused. Due to the large number of images in our dataset, uploading them individually would have been time-consuming, so we initially provided a compressed version for convenience.
>
> At this stage, we are unable to re-upload the images individually due to NeurIPS regulations, which prohibit modifying submission files during the review process. However, after the review period, we will upload each image separately and provide a detailed description of our PUO dataset and benchmark to ensure better accessibility and reproducibility.

---

> > ### Comment · Reviewer_BDQR · 2025-08-08
> >
> > The authors have offered solutions towards my concerns in the rebuttal, as my initial score is positive enough to show my support, I will keep my score.

---

> > > ### Author Response · Authors · 2025-08-08
> > > **Official Comment by Authors**
> > >
> > > Dear Reviewer BDQR,
> > >
> > > We are delighted to receive your positive feedback and truly appreciate your ongoing support. Your suggestions are indeed valuable to us.
> > >
> > > Yours sincerely,
> > >
> > > Authors of #1551

---

### Official Review · Reviewer_WD6h · 2025-07-01

**Rating:** 5
**Confidence:** 3

**Summary:**

This work constructs a dataset for panel understanding of household appliances. Panel operation is a highly practical function. This work defines F1 and Acc to evaluate the accuracy of a VLM in performing tasks from the perspective of panel description and function estimation. Subjective evaluation is introduced to retrospectively compare the performance of multiple VLMs.

**Dataset Code Accessibility:**

Yes

**Ethical Considerations:**

No, there are no or only very minor ethics concerns

**Limitations Weaknesses:**

I have three questions:
1) How is the ground truth obtained?
2) Are the subjective judgments of different people consistent for the same task?
3) For the misjudged data by VLM, an in-depth analysis of the causes can be provided: whether it is image quality interfering with recognition or UI design leading to ambiguity. Can VLM (fine tuned by this dataset) be applied to evaluate the rationality of panel UI design?

**Strengths Contributions:**

The contributions of this project are: 1) Dividing the task of panel understanding into panel description and function estimation, and providing corresponding prompts. 2) Collecting and annotating nearly 20,000 images. 3) Combining subjective experiments to verify the advantages of large models in this task

---

> ### Author Rebuttal · Authors · 2025-07-31
>
> We sincerely thank you for your careful reading of our paper and appreciate the valuable feedback in your comments. The insightful and constructive suggestions have enabled us to improve our work effectively.
>
> **1. [Re Question 1: Obtain Ground-truth.]**
>
> - Our annotation pipeline is as follows:
>
>   1. **Type and position annotation.**  This step is similar to annotations in detection tasks. Annotators are required to label all operable elements by indicating their types and bounding boxes. It is worth noting that bounding boxes not only capture the element itself but also surrounding contextual details like texts or symbols (as explained in L136 - L43).
>   2. **Text and icon annotation.**  This step borrows the OCR function and icon recognition features of existing VLMs to initially extract texts and icons in the bounding box. After that, a web tool is developed to correct mislabeling and missing labels for each box.
>   3. **Function annotation**. We develop a web tool and connect it to an LLM to estimate functions of elements. We use the following prompt: `This is an operable element on a {product}, and its type is {type}. The text written on or around this component is: '{text}'. The icon description associated with this component is: '{icon}'. Besides, its function is also related to '{other_prompts}'. Based on these details, what function might this element serve in operating the {product}? ` Normally, the LLM can correctly estimate the function of this element. Otherwise, annotators can write some words in the blank of `{other_prompts}` to make the LLM response as expected.
> - In step 1&2, the bounding box and its surrounding content correspond to the actual image context. In step 3, the function annotation is maintained consistently by the LLM's prompt — specific descriptions are based on the element’s context and human prompts (the `` {`other_prompt`} `` in the prompt).
>
> **2. [Re Question 2: Subjective Judgments of Different People.]**
>
> We understand your concern. We address this from the following aspects:
>
> - All participant gives nearly the same answer when given the same Question-Answer-GT triplets. To verify this, we also perform a human evaluation study on the **same** 10×3=210 Question-Answer-GT triplets to verify the consistency among 10 participants. The results are:
>
> ||description|function|planning|
> | ----------| :-----------: | :--------: | :--------: |
> |avg.std|0.0134|0.0043|0.0171|
> |inc.rate|0.0657|0.0014|0.0057|
>
> - By organizing all answers into a matrix $A = [a _ {ij}]^{10\times 70}$ with $a _ {ij}$ representing the response of the $i$-th participant to the $j$-th treplet, the above table demonstrates two metrics to measure the consistency:
>
>   1. "avg.std" represents the average standard deviation over all triplets in the given task, which can be expressed as: $\texttt{avg.std} = \frac 1 {70}\sum _ j \left[\frac 1 {10} \sum _ i(a _ {ij} -\frac 1 {10} \sum _ k a _ {kj})\right]$, in which the content within the square brackets represents the standard deviation of each triplet.
>   2. "inconsistent rate" refers to the number of instances where the current judgment $a_{ij}$ differs from the mode of responses to the same triplet: $\texttt{inc.rate} = \frac{1}{10 \times 70}\sum a _ {ij} \neq \mathrm{Mo} _ j(a _ {ij} ) $, where $ \mathrm{Mo} _ j(a _ {ij} ) $is the mode of the $j$-th triplet.
>
> According to the table, the inconsistency of participants' judgments on the same triplet is low, suggesting that the subjective judgments can be considered consistent.
>
> **3. [Re Question 3: In-depth Analysis of Misjudged Causes.]**
>
> Thanks for your feedback. We can analyze misjudged data in each task by sorting the corresponding scores and then checking these instances with a lower score. We find the following reasons may lead to misjudgment:
>
> - **Small button size and missing legend**: The button is too small to be recognized, since a small button also has a small legend to show its function. In the description task, small elements are normally ignored, while in the function estimation task, VLMs just produce hallucinatory outputs.
> - **Obscured perspective and implicit information**: Due to the perspective, some information cannot be directly captured but can be inferred from the surrounding information, such as the scale of a knob. However, for VLM, it only extracts relevant visual information to respond.
> - **Low image quality and ambiguity**: Image quality also has an impact. For some less clear images, the description task may intuitively ignore unclear elements, causing hallucinations in the grounding task and function estimation task. Besides, goal-based planning might create some buttons that do not actually exist.
>
> Additionally, there are some flaws in the grounding performance of VLMs. For the grounding task, the inferred coordinates are not of the element specified by the problem; for the function estimation task, it might misidentify the location, inferring a function that is not associated with the given coordinates of the element.
>
> As for UI design, PUO is more suitable for panel design in the real world. Current LLM can do it. Fortunately, our PPF keeps these abilities of LLM on the cloud side. Thus, one can use the light VLM to capture the distribution of a panel, and then send this information to a cloud-side LLM with the prompt "This is a description of a panel. Is its layout reasonable, and are there more suitable improvement plans?" Here is an example:
>
> > - ​`A button from (0.231, 0.584) to (0.33, 0.691). It has '' on it and looks like A fan icon with three blades. This is a button. Its possible function is to control the fan of the air conditioner. After pressing it, the fan can be turned on or the fan speed can be adjusted.`​
> > - ​`A button from (0.545, 0.588) to (0.641, 0.693). It has '' on it and looks like an upward-pointing triangle. This is a button. Its possible function is to increase the air conditioner's wind speed. After pressing it, the air conditioner's wind speed can be increased by one level upward.`​
> > - ​`A button from (0.069, 0.583) to (0.169, 0.693). It has '' on it and looks like A clock icon. This is a button. Its possible function is to set the timer function of the air conditioner. After pressing it, the timer duration of the air conditioner can be adjusted.`​
> > - ​`A button from (0.699, 0.588) to (0.796, 0.692). It has '' on it and looks like four small squares and a small circle. This is a button. Its possible function is to switch the air conditioner to a certain specific mode or function. When pressed, the corresponding mode or function can be activated.`​
> > - ​`A button from (0.389, 0.589) to (0.487, 0.695). It has '' on it and looks like A down arrow. This is a button that may be used to lower the temperature or wind speed of the air conditioner. When pressed, it may cause the set temperature to drop or the wind speed to weaken.`​
> > - ​`A button from (0.873, 0.588) to (0.939, 0.698). It has '' on it and looks like a power button. This is a button. Its possible function is to control the power supply of the air conditioner. Pressing it can turn the air conditioner on or off.`​
> >
> > The above list is a description of a panel. Is its layout reasonable, and are there more suitable improvement plans?
>
> LLMs can provide very useful advice on the UI layout. Due to the limitation of space in the rebuttal, we do not present the response of LLMs. However, by fine-tuning a good panel parser on our dataset, it can be used to evaluate and improve the panel UI design.

---

> > ### Comment · Reviewer_WD6h · 2025-08-06
> >
> > I appreciate the authors for sharing their insight on experimental results. I'm leaving my score unchanged.

---

> > > ### Author Response · Authors · 2025-08-07
> > > **Official Comment by Authors**
> > >
> > > Dear Reviewer WD6h，
> > >
> > > We are so glad to hear your positive comment! Thank you for your continuous support for the acceptance of our manuscript.
> > >
> > > Best regards,
> > >
> > > Authors of #1551

---

### Official Review · Reviewer_tdQn · 2025-07-03

**Rating:** 4
**Confidence:** 4

**Summary:**

This paper introduces PUO-Bench, a novel benchmark targeting the underexplored task of real-world Panel Understanding and Operation (PUO). The authors construct a large-scale vision-language dataset comprising over 19,000 annotated panel images and 430,000 instruction-following QA pairs, covering a wide range of appliances such as ovens, washing machines, and air conditioners. In addition, the paper presents a Privacy-Preserving Framework (PPF), which leverages edge-based visual parsing and cloud-based LLM reasoning to mitigate the privacy risks associated with uploading raw images. Extensive experiments compare zero-shot and fine-tuned VLMs, and demonstrate that the proposed dataset significantly enhances PUO-specific capabilities. The PPF approach is shown to be effective, achieving competitive performance while preserving user privacy.

**Dataset Code Accessibility:**

Yes

**Dataset Code Comments:**

The authors provide a dataset access link via HuggingFace, which is a standard and reliable platform for hosting research data.

**Ethical Considerations:**

No, there are no or only very minor ethics concerns

**Final Justification:**

A clear rebuttal is provided and a good work is done. I think it can be accepted.

**Limitations Weaknesses:**

- The connection between PUO task and the necessity of such a privacy-aware design is not strongly established. The paper lacks show that privacy is a dominant concern in realistic PUO applications. The jump from PUO task to a full edge-cloud split architecture feels abrupt and insufficiently motivated.
- All annotated elements are limited to "button" and "knob" types (Figure 4), which may oversimplify real-world panels that contain sliders, toggles, touchscreens, or dynamic displays. Additionally, the function annotations (Table 1 and Figure 5) are descriptive but highly abstract, and it's unclear whether they are consistently annotated across appliances or standardized. Without detailed annotation guidelines or inter-annotator agreement metrics, it is difficult to assess the dataset’s reliability as a benchmark.
- The proposed PPF framework relies on a lightweight detector (YOLO) and a small VLM at the edge, which limits the expressive power and reasoning capacity of the system. The edge-side model may cannot support more nuanced understanding or real-time feedback, and there's no latency or efficiency analysis to demonstrate PPF's deployability.

**Strengths Contributions:**

- PUO-Bench is the first large-scale dataset targeting real-world panel understanding and operation, offering over 19k images and 430k instruction-following QA pairs.
- The proposed PPF decouples edge-side visual parsing and cloud-side reasoning, effectively preventing raw image transmission and addressing real deployment concerns.
- The experiments compare zero-shot and fine-tuned VLMs, revealing the need for PUO-specific training. Results show that both the benchmark and PPF are effective.

---

> ### Author Rebuttal · Authors · 2025-07-31
>
> We deeply appreciate your thorough review of our manuscript and are grateful for the insightful feedback you provided. Your constructive comments have been instrumental in enhancing the quality of our work.
>
> **1. [Re Weakness 1: PPF Motivation.]**
>
> - We have provided the motivation for proposing PPF from L-50 to L-57 in the introduction section: when capturing images of panels, sometimes extra information is also included, such as the photos on the wall (mosaic part) shown in the bottom scene of Fig. 1. This poses a risk of exposing private data. Especially in the future, when the capturing device is an autonomous robot, private information may be uploaded to the cloud without the owner's knowledge. In order to better illustrate this risk, we provide the following evidence:
>
>   1. in Fig. 3(b), we show the proportion of pixels occupied by the panel, which shows the occupancy of panels accounting for less than 20%, which means 80% of the information is not related to the panel.
>   2. We used GPT to check each image and ask whether there is a risk of personal information or privacy leakage (such as location, personal phone numbers or names, photos of persons, etc) in it. The results from GPT indicate that approximately 3.67%  of the images have personal privacy leakage risks. This percentage is quite high in daily life.
> - Thus, to avoid this situation, we design a privacy-preserving framework that parses the image containing the panel locally, and then uses the cloud-side LLM to respond to the user's instructions accordingly, without posting private environmental information to any parts of LLM service provider. The following paragraph (L-58 to L-66) and Section 5 introduce how PPF addresses the concern in detail.
> - We appreciate your valuable comment, which helped us recognize the omission of the motivation for PPF in Section 5. In response, we will revise the manuscript by adding a paragraph at the beginning of Section 5 to explicitly explain the motivation and enhance the logical flow.
>
> **2. [Re Weakness 2-1: Overly Simplified Interactive Elements.]**
>
> We appreciate the reviewer’s concern. We have indeed considered this problem when collecting data, and our design choices were made with care.
>
> *2.1 Classification of Operable Elements.*
>
> We classified all operable elements into two categories: buttons and knobs, based on their interaction mechanisms:
>
> - **Buttons**: Represent binary switches that toggle between two states using the *same operation* (e.g., clicking or pressing).
> - **Knobs**: Represent multi-state controls that cycle through multiple positions or values by *turning to the desired position*.
>
> Accordingly, **sliders**, **toggles**, and **tap-based switches** are categorized as *knobs*, since they share similar operational characteristics (i.e., adjusting state by position rather than by discrete toggles).
>
> ​Examples:
>
> - `Dryer_ANIEKIN_1875_04680.jpg​`: This device has a slider to adjust wind speed with positions labeled as 'I', 'II', a snowflake icon, and 'O'. We annotate this element as a **knob** due to its similarity in usage (adjusting speed by position).`
> - ​`jd_Oven_Morphyrichards_018527cf.jpg`: This machine features a temperature slider ranging from 0℃ to 230℃ in 40℃ increments. We also annotate it as a **knob** because it is operated in the same way as a traditional rotary knob.
>
> Note that during annotation, we include the operable element along with surrounding texts and icons (see Lines 136 –143). This design allows the trained VLM to easily recognize the statefulness of each control by detecting multiple selectable states.
>
> *2.2 Touchscreens and Dynamic Displays.*
>
> For **touchscreens and dynamic displays**, we annotate each individual touchable element shown on the screen, similar to practices in web or UI understanding tasks.
>
> Examples in our dataset include:
>
> - ​`tb_Washing_Machine_Roborock_49cf27f4.jpg`​
> - ​`tb_Oven_Kaido_50a3472a.jpg`​
> - ​`jd_Air_Purifier_DYSON_8d2cb6e6.jpg`​
>
> These elements are operable through direct touch and are thus annotated as buttons.
>
> However, since touchscreen interactions are already well-tracked in existing GUI agent tasks, we deliberately limited the number of such samples in our dataset. Our PUO benchmark is primarily focused on physical operation panels, where real-world manipulation is required.
>
> **3. [Re Weakness 2-2: Highly Abstract  Function Annotations.]**
>
> - We understand your concern. This type of abstract description is not specifically targeted at the individual elements within a panel, and similar annotations also appear in GUI-related datasets [32, 38]. However, when combined with other contextual elements in a given scenario, such abstract descriptions can be effectively grounded into concrete operations by an LLM.
> - For example, in the right panel of Fig. 5, a button associated with the "air conditioner" and described as increasing the set temperature can be interpreted by the LLM as an operation to raise the indoor temperature when the user intends to do so.
> - More such cases can be found in the "Multi-Step Operation Planning" task of our PUO benchmark, where numerous examples demonstrate how abstract element functions are grounded into specific, actionable operations. Examples can be found in Fig.16 of our supplemental material.
>
> **4. [Re Weakness 2-3: Annotating Guideline.]**
>
> - Our annotation task is step-by-step, rather than having one annotator complete all five types of annotations. Thus, the consistency can be guaranteed in each annotation step. The annotion pipeline is as follows:
>
>   1. **Type and position annotation.**  This step is similar to annotations in detection tasks. Annotators are required to label all operatiable elements by indicating their types and bounding boxes. It is worth noting that bounding boxes not only capture the element itself but also surrounding contextual details like texts or symbols (as explained in L136 - L43).
>   2. **Text and icon annotation.**  This step borrow the OCR function and icon recognition features of existing VLMs to initially extract texts and icons in the bounding box. After that, a web tool is developped to help human annotator to correct mislabeling and missing labels for each box.
>   3. **Function annotation**. We develop a web tool and connect it to an LLM to estimate functions of elements. We use the following prompt: `This is a operatiable element on a {product}, and its type is {type}. The text written on or around this component is: '{text}'. The icon description associated with this component is: '{icon}'. Besides, its function is also related to '{other_prompts}'. Based on these details, what function might this element serve in operating the {product}? ` Normally, the LLM can correctly estimate the function of this element. Otherwise, annotators can write some words in the blank of `{other_prompts}` to make LLM response as expected.
> - In step 1&2, The bounding box and its surrounding content correspond to the actual image context. In step 3, the function annotation is maintained consistently by the LLM's prompt -- specific description are based on the element's context and human prompts (the `` {`other_prompt`} `` in the prompt).
> - Your comment made us realize that providing a more detailed annotation guideline could further enhance the clarity and reproducibility of our work. We will add this in the final version. We sincerely thank the reviewer again for the constructive suggestion.
>
> **5. [Re Weakness 3: PPF's Deployability.]**
>
> - We understand your concern that edge-side models are not good at reasoning due to their parameter limitations. However, PPF contains two parts: an edge-side parser and a cloud-side LLM. The latter is the major component for reasoning in PUO, while the edge-side parser only extracts operable elements on panels. Table 2 compares PPF with fine-tuned VLMs and zero-shot VLMs on the PUO benchmark, which shows that PPF surpasses them in grounding and function estimation.
> - As for the deployment of the edge-side model, we refer to [C1, C2, C3, C4] and test the latency of the edge-side parser. The edge-side hardware is a realmeNeo 7 running Android 15, equipped with a Dimensity 9300 processor, 16GB RAM, and 1TB ROM, and depolying VLM using Ollama.
>
>   - Using the input panel image with a resolution of $1536\times 768$, which is tokenized into 1540 tokens in QwenVL, we test the FPS of YOLO on our device and achieve 11.9 (frame/s), which equals 18326 (token/s).
>   - Besides, we obtain the speed of the deployed VLM on the edge side as follows:
>
> |Model|Input Speed (token/s)|Output Speed (token/s)|Storage (GB)|Memory (GB)|
> | :------------: | :---------------------: | :----------------------: | :------------: | :-----------: |
> |Qwen2.5VL-3B|1628.58|6.68|3.2|3.7|
>
> - However, we infer that the performance of edge-side models might still have untapped performance potential. The output speed of the edge-side model in [C1] can be greater than 30 token/s, nearly 5 times faster than our implementation. This suggests promising room for further optimization in future work.
>
> [C1] X.Lu, et al, "Bluelm-v-3b: Algorithm and system co-design for multimodal large language models on mobile devices," CVPR 2025.
>
> [C2] M.Abdin, et al, "Phi-4 technical report," arXiv, 2024.
>
> [C3] P.R. Guerrero, et al, "Efficient Deployment of Vision-Language Models on Mobile Devices: A Case Study on OnePlus 13R," arXiv, 2025.
>
> [C4] J.Xiao, et al. "Understanding Large Language Models in Your Pockets: Performance Study on COTS Mobile Devices," arXiv, 2024.

---

> > ### Comment · Reviewer_tdQn · 2025-08-03
> > **Good rebuttal files.**
> >
> > Authors address all my concerns. I will raise my score.

---

> > > ### Author Response · Authors · 2025-08-04
> > > **Official Comment by Authors**
> > >
> > > Dear Reviewer tdQn，
> > >
> > > Thank you for your time and effort in reviewing our paper. We are grateful for your feedback and pleased to hear your positive remarks!
> > >
> > > Best regards,
> > >
> > > Authors of #1551

---

### Author Response · Authors · 2025-08-01
**General Response to Reviewers**

### General Response

We sincerely appreciate the reviewers’ insightful comments and constructive feedback on our manuscript. We are pleased to receive positive recognition from most of the reviewers. In particular, we are delighted that the reviewers found our proposed real-world panel dataset and benchmark meaningful and informative (all reviewers), considered the proposed PPF method to be novel and effective (Reviewers tdQn, BDQR, X5m1), and regarded the experiments as convincing and impressive (all reviewers).

Based on the reviews, we provide below a general response to the commonly raised concerns, followed by individual responses to address each reviewer’s specific questions.

(1) **Regarding the experimental results, we have taken the following actions:**

- For Reviewer tdQn, we added an evaluation of the number of examples with potential privacy leakage risks.

- For Reviewers tdQn, BDQR, and X5m1, we conducted additional experiments analyzing the speed and memory usage for edge-side deployment.

- For Reviewer WD6h, we supplemented the manuscript with a human consistency verification experiment.

- For Reviewer X5m1, we provided additional experiments evaluating generalization to unseen panels.

(2) **In response to questions about the motivation, benchmark design, and technical details, we have made the following clarifications:**

- For Reviewer tdQn, we clarified the motivation behind PPF, the rationale for element design choices, and the method’s deployability.

- For Reviewer WD6h, we clarified the pipeline for obtaining ground truth, addressed concerns regarding subjective judgment variability, and provided an in-depth analysis of misjudged data.

- For Reviewer BDQR, we addressed questions related to deployment on mobile devices.

- For Reviewer X5m1, we addressed concerns about domain coverage, run-time performance, privacy assumptions, generalization ability, and potential regulatory implications.

(3) **Regarding the valuable suggestions on presentation and organization from all reviewers**, we will incorporate them and make the necessary revisions in the final version of the paper.

Once again, we thank you for your thoughtful and constructive suggestions -- they have greatly helped us improve the quality and clarity of the paper. Over the next week, we would be glad to post extended replies on the forum. Please don't hesitate to let us know if you would like any further clarifications or if there are minor points we can address. We'd be happy to provide additional information to help convey the merits of our work. We truly appreciate your feedback.

Yours sincerely,
Authors of #1551

---

### Author Response · Authors · 2025-08-08
**General Response to Reviewers**

### Gratitude for Your Feedback & Inquiry on Further Questions (Deadline Approaching)

Dear Reviewers,​

Thank you all for your insightful responses to our rebuttal. We are pleased to know that we have addressed all questions raised so far.​ We are particularly delighted to receive positive assessments from **all reviewers**, including reviewers **tdQn** and **X5m1** affirming our rebuttal and considering a score increase, as well as **WD6h** and **BDQR** giving us highly positive feedback right from the start.  All your positive feedback not only acknowledges our effort during rebuttal time, but also greatly helps us improve the manuscript.

As the discussion DDL is approaching, please let us know if there are any further questions or concerns—we will promptly provide additional clarifications. Again, thank you for your recognition and endeavor in improving our paper.​

Best regards,​

Authors of #1551

---

### Decision · Program_Chairs · 2025-09-18

**Decision:**

Accept (poster)

**Comment:**

The reviewers’ scores were BA, A, A, and A, showing strong overall support for acceptance. The paper’s key contributions are the introduction of PUO-Bench, the first large-scale dataset for panel understanding and operation, and the proposed PPF framework, which decouples edge-side parsing from cloud-side reasoning to address deployment and privacy concerns. Reviewers acknowledged the dataset’s scale, the novelty of the framework, and the convincing experiments. Some concerns were raised, including whether privacy is truly a dominant issue for PUO tasks, the limited scope of annotated panel elements, the lack of annotation reliability analysis, and questions around the expressive power and deployability of the edge-side model. The rebuttal effectively addressed these points by providing additional analyses (e.g., privacy risk evaluation, edge-side efficiency, human consistency verification, and generalization experiments), as well as clarifications on motivation, annotation process, and deployment details. Given the strong positive reception and the authors’ thorough responses, the AC encourages the authors to integrate these revisions and clarifications into the final version.